

Aerosol distribution in the northern Gulf of Guinea: local anthropogenic
sources, long-range transport and the role of coastal shallow
circulations
Cyrille Flamant[1], Adrien Deroubaix[1,2], Patrick Chazette[3], Joel Brito[4], Marco Gaetani[1],
Peter Knippertz[5], Andreas H. Fink[5], Gaëlle de Coetlogon[1], Laurent Menut[2], Aurélie
Colomb[4], Cyrielle Denjean[6], Remi Meynadier[1], Philip Rosenberg[7], Regis Dupuy[4], Alfons
Schwarzenboeck[4] and Julien Totems[3]
[1]Laboratoire Atmosphères Milieux Observations Spatiales, Sorbonne Université,
Université Paris-Saclay and CNRS, Paris, France
[2]Laboratoire de Météorologie Dynamique, Ecole Polytechnique, IPSL Research
University, Ecole Normale Supérieure, Université Paris-Saclay, Sorbonne Université,
CNRS, Palaiseau, France
[3]Laboratoire des Sciences du Climat et de l'Environnement, CEA, CNRS, Université
Paris-Saclay, Gif-sur-Yvette, France
[4]Laboratoire de Météorologie Physique, Université Clermont Auvergne, CNRS,
Clermont-Ferrand, France
[5]Institute of Meteorology and Climate Research, Karlsruhe Institute of Technology,
Karlsruhe, Germany
[6]Centre National de Recherches Météorologiques, Météo-France and CNRS,
Toulouse, France
[7]Institute of Climate and Atmospheric Science, School of Earth and Environment,
University of Leeds, Leeds, United Kingdom



**Abstract**
*The complex vertical distribution of aerosols over coastal southern West Africa (SWA)*
*is investigated using airborne observations and numerical simulations. Observations*
*were gathered on 2 July 2016 offshore of Ghana and Togo, during the field phase of*
*the Dynamics-Aerosol-Chemistry-Cloud Interactions in West Africa project. The*
*aerosol loading in the lower troposphere includes emissions from coastal cities*
*(Accra, Lomé, Cotonou and Lagos) as well as biomass burning aerosol and dust*
*associated with long-range transport from Central Africa and the Sahara,*
*respectively. Our results indicate that the aerosol distribution is impacted by*
*subsidence associated with zonal and meridional regional scale overturning*
*circulations associated with the land-sea surface temperature contrast and*
*orography over Ghana and Togo. Numerical tracer release experiments highlight the*
*dominance of aged emissions from Accra on the observed pollution plume loadings*
*over the ocean. The contribution of aged emission from Lomé and Cotonou is also*
*evident above the marine boundary layer. Lagos emissions do not play a role for the*
*area west of Cotonou. The tracer plume does not extend very far south over the*
*ocean (i.e. less than 100 km from Accra), mostly because emissions are transported*
*northeastward near the surface over land and westward above the marine*
*atmospheric boundary layer. The latter is possible due to interactions between the*
*monsoon flow, complex terrain and land-sea breeze systems, which support the*
*vertical mixing of the urban pollution. This work sheds light on the complex – and to*
*date undocumented – mechanisms by which coastal shallow circulations distribute*
*atmospheric pollutants over the densely populated SWA region.*



1. Introduction

Aerosol-cloud-climate interactions play a fundamental role in radiative balance and energy redistribution in the tropics. Aerosol particles from natural and anthropogenic origins can serve as cloud condensation nuclei (Haywood and Boucher, 2000; Carslaw et al., 2010) and interact with solar and terrestrial radiation through absorption and scattering.

The atmosphere over southern West Africa (SWA) is a complex mix of local emissions (vegetation, traffic, domestic and waste fires, power plants, oil and gas rigs, ships) and remote sources (dust from the north and wild-fire related biomass burning aerosols from Central Africa) (Knippertz et al., 2015a, Brito et al., 2018). In order to enhance our understanding of aerosol-cloud-climate interactions in SWA, it is of paramount importance to better characterize the composition and vertical distribution of the aerosol load over the eastern tropical Atlantic. This is particularly vital, since SWA is currently experiencing major economic and population growths (Liousse et al., 2014), and is projected to host several megacities (cities with over 10 million inhabitants) by the middle of the 21st century (World Urbanization Prospect, 2015). This will likely boost anthropogenic emissions to unprecedented levels and imply profound impacts on population health (Lelieveld et al., 2015), on the radiative budget over SWA and also on the West African Monsoon (WAM) system (Knippertz et al., 2015b). This will also add to the dust and biomass burning aerosol related perturbations already evidenced for the precipitation in the area (e.g. Huang et al., 2009). Likewise, urban pollution may also affect surface-atmosphere interactions and associated lower tropospheric dynamics over SWA as for instance dust over the



tropical Atlantic (e.g. Evan et al., 2009) or biomass burning aerosols over Amazonia
(Zhang et al., 2008, 2009).

One of the aims of the EU-funded project Dynamics-Aerosol-Chemistry-Cloud
Interactions in West Africa (DACCIWA, Knippertz et al., 2015b) is to understand the
influence of atmospheric dynamics on the spatial distribution of both anthropogenic
and natural aerosols over SWA. One particularly important aspect is the fate of
anthropogenic aerosols emitted at the coast as they are being transported away
from the source. In addition, DACCIWA aims at assessing the impact of this complex
atmospheric composition on the health of humans and ecosystems.

Urban aerosols are mostly transported with the southwesterly monsoon flow below
700 hPa (e.g. Deroubaix et al., 2018). They may also reach the nearby ocean as the
result of complex dynamical interactions between the monsoon flow, the
northeasterly flow from the Sahel above and the interactions with the atmospheric
boundary layer (ABL) over the continent coupling the two layers when it is fully
developed during daytime. This is because, as opposed to the marine ABL, the
continental ABL exhibits a strong diurnal cycle (e.g. Parker et al., 2005; Lothon et al.,
2008; Kalthoff et al., 2018). On hot, cloud-free summer days, land-sea breeze systems
can develop at the coast (in conditions of moderate background monsoon flow,
Parker et al., 2017), which contribute to the transport of pollutants emitted along the
urbanized coastal strip of SWA.

The main objective of the present study is to understand how the lower tropospheric
circulation over SWA shapes the urban pollution plume emitted from coastal cities
such as Accra, Lomé, Cotonou and Lagos, both over the Gulf of Guinea and inland.



Here, we take advantage of the airborne measurements acquired during the
DACCIWA field campaign (June–July 2016, Flamant et al., 2018) as part of the
European Facility for Airborne Research (EUFAR) funded Observing the Low-level
Atmospheric Circulation in the Tropical Atlantic (OLACTA) project to assess the
characteristics of different aerosol layers observed over the Gulf of Guinea. To study
the role of atmospheric dynamics on aerosol spatial distribution, we use a unique
combination of airborne observations from the 2 July 2016, space-borne observations
and finally high-resolution simulations performed using the Weather and Research
Forecast (WRF) and CHIMERE models.

The airborne and space-borne data used in this paper are presented in Section 2,
whereas the simulations are detailed in Section 3. Section 4 presents the synoptic
situation and airborne operations over SWA on 2 July 2016. Atmospheric composition
over the Gulf of Guinea as observed from aircraft in situ and remote sensing data is
discussed in Section 5. Insights into the distribution of anthropogenic aerosols from
tracer experiments are presented in Section 6 and long-range transport of aerosols
related to regional-scale dynamics is described in Section 7. The influence of lower-
tropospheric overturning circulations induced by the land-sea surface temperature
gradient on the vertical distribution of aerosols over SWA is discussed in Section 8. In
Section 9, we summarize and conclude.

2.  Data


2.1 Airborne observations




During the DACCIWA field campaign, airborne operations on the afternoon of 2 July
2016 were conducted with the ATR 42 of the Service des Avions Français Instrumentés
pour la Recherche en Environnement (SAFIRE) over the Gulf of Guinea (**Figure 1**). The
afternoon flight was carried out in the framework of the EUFAR OLACTA project
(Flamant et al., 2018). The aircraft was equipped with in situ dynamical and
thermodynamical probes (yielding mean and turbulent variables), as well as in situ
aerosol and cloud probes, and gas phase chemistry instruments. It also carried
several radiometers (upward and downward looking pyranometers and
pyrgeometers) as well as the Ultraviolet Lidar for Canopy Experiment (ULICE, Shang
and Chazette, 2014). **Table 1** summarizes the instruments used in this study (see the
Supplement of Flamant et al., 2018 for the complete ATR 42 payload during the field
campaign).

2.1.1 ULICE observations

The ULICE system was specifically designed to monitor the aerosol distribution in the
lower troposphere. During the DACCIWA field campaign, ULICE was pointing to the
nadir. The system's nominal temporal and along-line-of-sight resolutions are 100 Hz
and 15 m, respectively. In the present study, we use lidar-derived profiles of aerosol-
related properties averaged over 1000 laser shots (~10 s sampling).

The ULICE receiver implements two channels for the detection of the elastic
backscatter from the atmosphere in the parallel and perpendicular polarization
planes relative to the linear polarization of the emitted light. The design and the
calculations to retrieve the depolarization properties are explained in Chazette et al.
(2012). Using co- and cross-polarization channels, the lidar allows identifying non-



spherical particles in the atmosphere such as dust. The overlap factor is nearly
identical for the two polarized channels, thereby permitting the assessment of the
volume depolarization ratio (VDR) very close to the aircraft (~150 m).

Lidar-derived extinction coefficient profiles (as well as other optical properties) are
generally retrieved from so-called inversion procedures as abundantly described in
the literature (e.g. Chazette et al., 2012). During the DACCIWA field campaign the
lack of adequate observations did not allow us to perform proper retrievals of
aerosol optical properties using such procedures. Hence, in the following we only use
the apparent scattering ratio (ASR, the ratio of the total apparent backscatter
coefficient to the molecular apparent backscatter coefficient denoted $R_{app}$) and
the VDR. Details are given in **Appendix A**, together with the characteristics of the
lidar system.

Generally speaking, the VDR values observed during the flight are not very high and
absolute values may be subject to biases. Nevertheless, relative fluctuations of VDR
are accurately measured and useful as indicators of changes in aerosol properties.

2.1.2 Aerosol and gas phase chemistry measurements


For this study, we focus on available observations that can provide insights into the
origin of the aerosol distribution over coastal SWA, namely biomass burning aerosols,
dust and urban pollution:
- **Biomass burning aerosols**: identification was conducted at times of enhanced

ozone ($O_3$) and carbon monoxide (CO) mixing ratios as well as aerosol

parameters such as light absorption/extinction and number concentration.





- **Urban pollution**: the main tracers used were CO, nitrogen oxide (NOx) and
total (>10 nm) particle number concentrations;
- **Terrigenous aerosols (dust)**: layers were identified at times of enhanced
aerosol parameters (particularly super micron aerosols), in complement to the
lidar-derived VDR observations and not followed by CO or $O_3$ enhancements
(mostly associated with biomass burning here).

In addition, absorption Angstrom exponent (AAE) measurements are used to
distinguish urban air pollution from biomass burning smoke (Clarke et al., 2007) and
mineral dust (Collaud Coen et al., 2004). In general the AAE values for carbonaceous
particles are ~1 for urban pollution, between 1.5 and 2 for biomass smoke and
around 3 for dust (Bergstrom et al., 2007).

The Particle Soot Absorption Photometer (PSAP, model PSAP3L) measures the aerosol
optical absorption coefficient at three wavelengths (467, 530 and 660 nm) with a
sampling time of 10 s. The data were corrected for multiple scattering and
shadowing effects according to Bond et al. (1999) and Müller et al. (2009). Data with
filter transmission under 0.7 are removed as corrections are not applicable.
Furthermore, PSAP measurements were used to compute the AAE. The particle
extinction coefficient is measured with a cavity attenuated phase shift particle light
extinction monitor (CAPS-PMex, Aerodyne Research) operated at the wavelength of
530 nm. Data were processed with a time resolution of 1 s. An integrated
nephelometer (Ecotech, model Aurora 3000) provided aerosol light scattering at
three wavelenghts (450, 550 and 700 nm), which was used to correct for the impact
of aerosol scattering based on the correction scheme by Anderson and Ogren
(1998) and using correction factors obtained by Müller et al. (2011) without a



submicron size cut-off. The nephelometer was calibrated with particle-free air and
high-purity $CO_2$ prior to and after the campaign.

Prior to the campaign, the CAPS data were evaluated against the combination of
the nephelometer and the PSAP measurements. The instrument intercomparison has
been performed with purely scattering ammonium sulfate particles and with strongly
absorbing black carbon particles. Both types of aerosols were generated by
nebulizing a solution of the respective substances and size-selected using a
Differential Mobility Analyzer. For instrument intercomparison purposes, the extinction
coefficient from the nephelometer and PSAP was adjusted to that for 530 nm by
using the scattering and absorption Angstrom exponent. The instrument evaluation
showed an excellent accuracy of the CAPS measurements by comparison to the
combination of nephelometer and PSAP measurements. The level of uncertainty
obtained for the test aerosol was beyond the upper limit of the CAPS uncertainty
which was estimated to be +-3% according to Massoli et al. (2010).

Total particle concentration for particle diameters above 10 nm ($N_{10}$) are made using
a Condensational Particle Counter (CPC, model MARIE built by University of Mainz),
calibrated prior to the experiment (sampling time 1 Hz).  Aerosol optical size in the
range 0.25–25 μm is measured using an Optical Particle Counter (OPC, model 1.109
from GRIMM Technologies) in 32 channels, with a 6 s sampling rate. Particulate
matter number concentrations for size ranges smaller than 1 μm, between 1 and 2.5
μm and between 2.5 and 10 μm are computed from the OPC, and are referred to
$N_{PM1}$, $N_{PM2.5}$ and $N_{PM10}$ respectively, in the following. The GRIMM OPC was calibrated
with size-standard particles prior and after the field campaign.



Sampling with all the above mentioned instruments is achieved through the
Community Aerosol Inlet of the ATR 42.

Regarding gas phase chemistry, we make use of an $O_3$ analyzer and a NOx analyzer
from Thermo Environmental Instruments (TEI Model 49 and TEI 42CTL, respectively).
Carbon monoxide (CO) measurements are performed using the near-infrared cavity
ring-down spectroscopy technique (G2401, Picarro Inc., Santa Clara, CA, USA), with
a time resolution of 5 s.

All in-cloud measurements are removed from the data shown here.

2.2 Space-borne observations


The Spinning Enhanced Visible and Infra-Red Imager (SEVIRI), onboard Meteosat
Second Generation (MSG), measures aerosol optical depth (AOD) with spatial and
temporal resolutions of 10 km and 15 min, respectively (Bennouna et al., 2009). We
use the operational version 1.04 of the AOD product at 550 nm, downloaded from
the ICARE data service center (http://www.icare.univ-lille1.fr/).

The Moderate Resolution Imaging Spectroradiometer (MODIS, Salmonson et al., 1989;
King et al., 1992) flies aboard the polar-orbiting platforms Aqua and Terra. Terra
crosses the Equator from north to south in the morning (~1030 local time), whereas
Aqua crosses from south to north during the afternoon (~1330 local time). They
provide a complete coverage of the Earth surface in one to two days with a
resolution between 250 and 1000 m, depending on the spectral band. In the
following, we use MODIS-derived level 2 AODs at 550 nm from both Terra and Aqua.

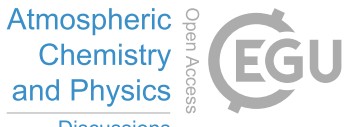

Level 2 products are provided as granules with a spatial resolution of 10 km at nadir.
The standard deviation on the AOD retrieval (Remer et al., 2005) over land (ocean) is
0.15±0.05xAOD (0.05±0.03xAOD). We also use level 3 daily sea surface temperature
(SST) data derived from the 11 µm thermal infrared band available at 9.26 km spatial
resolution for daytime passes (Werdell et al., 2013).

The hourly land surface temperature products from the Copernicus Global Land
Service (https://land.copernicus.eu/global/products/lst) used in this study are
available at 5 km spatial resolution. The radiative skin temperature of the land
surface is estimated from the infrared spectral channels of sensors onboard a
constellation of geostationary satellites (among which SEVIRI on MSG). Its estimation
further depends on the surface albedo, the vegetation cover and the soil moisture.

The Cloud-Aerosol LIdar with Orthogonal Polarization (CALIOP) flies onboard the
Cloud-Aerosol Lidar Pathfinder Satellite Observation (CALIPSO), following a similar
polar orbit than Aqua within the A-train constellation. In this work, we use CALIOP
level-2 data (version 4.10) below 8 km above mean sea level (amsl; https://www-
calipso.larc.nasa.gov/products/). Details on the CALIOP instrument, data acquisition
and science products are given by Winker et al. (2007). We mainly consider the
aerosol typing, which was corrected in version 4.10, as described in Burton et al.

(2015).


2.3 Radiosounding network

During the DACCIWA field campaign, the upper air network was successfully
augmented in June and July 2016 to a spatial density unprecedented for SWA (see




Flamant et al., 2018). In this study, we use radiosounding data from meteorological
balloons launched in Abidjan, Accra and Cotonou in the afternoon of 2 July (see
**Figure 1**). The management of soundings at Abidjan and Cotonou was
subcontracted to a private company, while the Ghana Meteorological Agency took
care of the soundings in Accra. The Karlsruhe Institute of Technology was instrumental
in the Ghana sounding and staff from the Agence pour la Sécurité de la Navigation
Aérienne en Afrique et à Madagascar helped with the Abidjan and Cotonou
soundings.

3. Models and simulations

3.1 ECMWF operational analyses & CAMS forecasts

For the investigation of atmospheric dynamics at the regional scale, we use
operational analyses from the Integrated Forecasting System (IFS, a global data
assimilation and forecasting system) developed by the European Centre for Medium-
Range Weather Forecasts (ECMWF). The analyses presented in this paper are
associated with IFS model cycle CY41r2. The original $T_{co}1279$ (O1280) resolution of the
operational analysis was transformed onto a 0.125° regular latitude-longitude grid.
Long-range transport of biomass burning and dust laden air masses transported over
the Gulf of Guinea are monitored with respective optical depths at 550 nm
calculated from the ECMWF Copernicus Atmosphere Monitoring Service-Integrated
Forecasting System (CAMS-IFS; Flemming et al., 2015) available at a resolution of 0.4°.

3.2 WRF and CHIMERE simulations



The WRF model (version v3.7.1, Shamarock and Klemp, 2008) and the CHIMERE
chemistry-transport model (2017 version, Mailler et al., 2017) are used in this study.
WRF calculates meteorological fields that are then used by CHIMERE to (i) conduct
tracer experiments and (ii) compute backplumes. WRF and CHIMERE simulations are
performed on common domains. For the period 30 June—3 July 2016, two simulations
are conducted to provide insights into the airborne observations: a simulation with a
10-km mesh size in a domain extending from 1°S to 14°N and from 11°W to 11°E
(larger than the domain shown in **Figure 1a**) and a simulation with a 2-km mesh size in
a domain extending from 2.8°N to 9.3°N and from 2.8°W to 3.3°E (**Figure 1a**). The 10-
km WRF simulation uses National Center for Environmental Prediction (NCEP) Final
global analyses as initial and boundary conditions. NCEP Real-Time Global SSTs
(Thiébaux et al., 2003) are used as lower boundary conditions over the ocean. The
meteorological initial and boundary conditions for the 2-km WRF simulation are
provided by the 10-km WRF run. The simulations are carried out using 32 vertical
sigma-pressure levels from the surface to 50 hPa, with 6 to 8 levels in the ABL for WRF
and with 20 levels from the surface to 300hPa for CHIMERE. This model configuration is
the same as described in Deroubaix et al. (2018).

The representation of the atmospheric dynamics in the 2-km simulation was verified
against dynamical and thermodynamical observations from both aircraft (**Figure S1**)
and the DACCIWA radiosounding network from Accra and Cotonou (**Figure S2**),
yielding satisfactory results.

3.2.1   Tracer experiments






A series of numerical tracer experiments were conducted to aid interpreting airborne
observations, particularly by separating (locally emitted) urban pollution from long-
range transported aerosol plumes. Passive tracers were set to be released from four
major coastal cities: Accra (Ghana, 5.60°N, 0.19°W), Lomé (Togo, 6.17°N, 1.23°E),
Cotonou (Benin, 6.36°N, 2.38°E) and Lagos (Nigeria, 6.49°N, 3.36°E). We conducted
two sets of experiments: one for which emissions from the cities are identical (TRA_I,
with "I" standing for "identical") and one for which the emissions are different and
proportional to the size of the population (TRA_D, with "D" standing for "different"),
based on the World Urbanization Prospect report (2015). In the latter case, emissions
from Lomé, Accra and Lagos are scaled to Cotonou emissions (1.8, 3 and 13 times,
respectively). Tracers are emitted in the lowest level of the model (below 10 m
altitude) during the period of interest: in experiences TRA_D1 and TRA_I1, tracers are
emitted continuously on 1 and 2 July, while in experiences TRA_D2 and TRA_D3,
tracer emissions only occur on 1 July and 2 July, respectively. Emissions take place in
a 2 km x 2 km mesh for each city. For the sake of simplicity, emissions are constant in
time and thus do not have a diurnal cycle. Tracer concentrations in the atmosphere
are then shown in arbitrary units (a.u.) and colored according to the city: blue for
Accra, green for Lomé and red for Cotonou. By design, the lifetime of the tracers
emitted at a given time from any of the considered cities is 48h. After that time,
tracers have either moved out of the domain or their concentration is set to zero.

3.2.2    Backplumes


Backplumes (or back trajectory ensembles) are computed according to Mailler et al.
(2016), using a dedicated regional CHIMERE simulation with a mesh size of 30 km,
covering the whole of Africa. For this study, 50 tracers are released at the same time



for selected locations along the ATR 42 flight trajectory, where large aerosol contents
are observed: (i) the southernmost part of the flight (2.0°W, 4.5°N) and (ii) the
northernmost part of the flight (1.0°E, 5.5°N). For both locations, backplumes are
launched at 2500 m above sea level on 2 July 2016 at 17:00 UTC. Very similar results
are obtained for both backplumes. Hence, in the following we shall only show results
from the backplume released from the northernmost location.

4.  Synoptic situation and airborne operations on 2 July 2016


The entire DACCIWA aircraft campaign took place during WAM post-onset
conditions (Knippertz et al., 2017), i.e. after the migration of the climatological
precipitation maximum from the coast to the Sahel, with the monsoon flow being
well established over SWA. The campaign also took place after the onset of the
Atlantic Cold Tongue as evident in Figure 3 of Knippertz et al. (2017), which also
highlights that the coastal upwelling started progressively building up around 27 June

2016.


In the period spanning from 29 June to 5 July 2016, the major weather disturbances
over SWA are associated with African Easterly Waves traveling along a well-
organized African Easterly Jet (AEJ). A cyclonic center propagating to the south of
the AEJ (identified from ECMWF 850 hPa streamline charts, not shown) originated
from eastern Nigeria on 29 June, sweeping through SWA during the following days.

On 2 July 2016, the cyclonic center is located at the coast of Sierra Leone (see
disturbance labelled "F" in Fig. 14 of Knippertz et al., 2017). The monsoonal winds are
almost southerly over the Gulf of Guinea (south of 4°N) and progressively veer to



southwesterly farther north and over the continent (**Figure S3a**). In the mid-
troposphere, SWA is under the influence of easterly flow conditions (**Figure S3b**). West
of 5°E, the AEJ is located over the Sahel and is intensified along its northern boundary
by a strong Saharan high located over Libya. The AEJ maximum is seen off the coast
of Senegal.

The region of interest experiences high insolation on 1 July with temperatures in the
30s °C across SWA and widespread low-level clouds dissolving rapidly in the course of
the morning. On 2 July, there is a clear indication of land-sea breeze clouds in the
high-resolution SEVIRI image at 1200 UTC (**Figure 2a**) with relatively cloud-free
conditions over the ocean, where the ATR 42 flew later on. The land-sea breeze front
is seen in-land to follow the coastline from western Ghana to western Nigeria. The
front is observed to move farther in-land until 1500 UTC (**Figure 2b**) with shallow
convective cells forming along it. Farther south the area is free of low-level clouds
(both over land and ocean). Oceanic convection occurred offshore on the previous
day and mesoscale convective systems were present over north-central Nigeria in
the morning of 2 July. Satellite images show both oceanic and inland convection to
be decaying by midday (**Figure 2a**).

On 2 July, the ATR 42 aircraft took off from Lomé at 1445 UTC (*NB: UTC equals local*
*time in July in Togo*) and headed towards the ocean, flying almost parallel to the
Ghana coastline (**Figure 1a**) at low level (in the marine ABL). Before reaching the
Cape Three Points (close to the border between Ghana and Ivory Coast), the ATR 42
changed direction and headed south. Upon reaching its southernmost position
(~3°N), the ATR 42 turned around and climbed to 3200 m amsl and finally headed
back to Lomé at that level. On the way back, the aircraft changed heading around



1653 UTC to fly along the coast prior to landing. The ATR 42 passed the longitude of
Accra at 1729 UTC and landed in Lomé at 1807 UTC. The high-level flight back
allowed mapping out the vertical distribution of aerosols and clouds using the lidar
ULICE. In situ aerosol and gas phase chemistry measurements will be used in the
following to characterize the composition of aerosols and related air masses
sampled with the lidar, particularly during the ascent over the ocean (between 1633
and 1647 UTC), the elevated leveled run and the descent towards the Lomé airport
(between 1753 and 1807 UTC).

5. Atmospheric composition over the Gulf of Guinea and the link with lower

tropospheric circulation


**Figure 3** shows ULICE-derived ASR and VDR cross-sections acquired between 1640
UTC and 1800 UTC, including data gathered during the aircraft ascent over the
ocean and descent in the vicinity of the coast. It is worth noting that most of the lidar
data shown in **Figure 3** were acquired while the aircraft was flying along the
coastline (from 1653 UTC on). Wind measurements from the Abidjan, Accra and
Cotonou soundings as well as from the ATR 42 sounding over the ocean clearly show
that above 1.2 km amsl the flow is easterly over the region of aircraft operation
(**Figure 4**). Given that the heading of the aircraft along this elevated leg is 65°, the
lidar "curtains" above 1.2 km amsl in **Figure 3** are mapping out aerosol layers that are
transported westward (with the ATR 42 flying against the mean flow).

Several outstanding features are highlighted in **Figure 3**. Generally few clouds were
encountered along the flight track (they appear in dark red colors). Exceptions are
the low-level clouds at the top of the marine ABL with a base around 500 m amsl to



the west of the track between 1655 and 1702 UTC (**Figure 3a**). The vertical extension
and the number of the cumulus clouds topping the marine ABL decreases towards
the east. This shoaling of the marine ABL is likely ascribed to the increasing trajectory
length of near-surface parcels over the cold coastal waters (as the aircraft flies over
the coastal upwelling region). Near Lomé, the top of the marine ABL can only be
identified from the higher ASR values reflecting the impact of high relative humidity
on the scattering properties of the marine aerosols (**Figure 3a**). An isolated deeper
convective cloud is observed before 1648 UTC between 2 and 2.5 km, which is also
sampled in situ by the ATR 42 cloud probes. The top of the cloud is likely connected
to a temperature inversion observed during the aircraft ascent over the ocean (not
shown). High lidar-derived ASRs are observed near the marine ABL top and to some
extent in the mixed layer (**Figure 3a**). The ASR-enhanced layers do not show on the
VDR plot, Possibly because they are related to the presence sea-salt aerosols which
are spherical particles that do not depolarize the backscattered lidar signal.
However, the high ASR values could also be related to the advection of biomass
burning aerosols from the south in the marine ABL (e.g. Menut et al., 2018) as
suggested by the relatively high CO and extinction coefficient values observed in
the ABL over the ocean (110 ppb and 50 Mm$^{-1}$, respectively) in Figure 4c and 4e.
Biomass burning aerosols are also generally associated with low VDR values.

In addition to clouds and marine ABL aerosols, several distinct aerosol features in the
free troposphere stand out from the lidar plot:
• **Features A and B** correspond to plumes with high values of ASR (larger than 1.2)
and VDR (larger than 0.8%) observed near the coast between the surface and 0.5
km amsl and between 0.5 and 1.5 km amsl, respectively, during the aircraft
descent towards Lomé. According to the aircraft in situ observations, feature B is



located in a strong wind shear environment at the top (~600 m) of the ABL (**Figure**
**4**) with its upper part being located in the easterly flow, while feature A is
associated with a south-southwesterly flow. This sheared environment likely
explains the slanted structure of the aerosol plume associated with feature B.
• **Feature C** is an intermediate aerosol layer characterized by VDR values lower than
those for feature B, suggesting more spherical (possibly more aged pollution)
aerosols. This feature is bounded by much lower VDR values, especially above,
while being associated with higher ASR values than its immediate environment.
This feature is slanted between Lomé and the deeper isolated cloud. The layer
thickness is larger near Lomé than over the more remote ocean, leading to a less
slanted layer top. This layer has also been sampled in situ by the ATR 42 during its
ascent over the ocean. It is characterized by VDRs on the order of 0.7%. Based on
the aircraft sounding data, it appears that this layer is mostly advected with the
easterly flow above 1.2 km amsl (**Figure 4**).
• **Feature D** is an elevated aerosol layer observed at the level of the aircraft (i.e. at
3200 m amsl) in the vicinity of Lomé, which was also sampled in situ by the ATR 42.
This layer is separated from feature B by a ~500 m deep layer of non-depolarizing
aerosols (very low VDRs). The base of this layer exhibits a slanting similar to the one
observed for the top of the intermediate aerosol layer (feature B). Large VDRs are
found in the core of this feature (> 1.2%). It appears that this layer is also advected
with the easterly flow above 1.2 km amsl.
• **Feature E** is also an elevated aerosol layer, but observed farther south over the
ocean and in the vicinity of the isolated deeper cloud. It is characterized by large
ASR values but low VDR values (suggesting the presence of low-depolarizing
aerosols).



Given the distance of the oceanic profile to the coast (~100 km), we consider the
oceanic (ascending) profile as representative of background aerosol/gas phase
conditions upstream of coastal SWA. Using this profile as reference, we have
analyzed the characteristics of the aerosol plume sampled with the ATR 42 (both in
situ and remotely) during the aircraft descent over Lomé. The most significant
differences between the ATR 42 observations acquired during the oceanic profile
and the profile over Lomé are found below 1.7 km amsl (**Figure 5**) and are
associated with features A and B.

ATR 42 observations associated with feature A (below 0.5 km amsl) show increases in
$NO_x$, CO and PM1 aerosol concentrations (**Figure 5a, c, f**, respectively) as well as
extinction coefficient (**Figure 5e**), together with an $O_3$ concentration reduction
(**Figure 5b**). No CPC-derived aerosol concentrations are available below 0.5 km
amsl. The few PSAP measurements made around 0.5 km amsl during the descent
yield an AAE value around ~1 (**Figure 5g**). Furthermore, the ratio of $O_3$ to CO
concentrations is on the order of 0.15 (not shown). These are solid indications that the
ATR 42 sampled a fresh urban anthropogenic plume near Lomé (Brito et al., 2018),
advected with the south-southwesterly monsoon flow (the ATR 42 being downstream
of Lomé then).

ATR 42 observations associated with feature B (between 0.5 and 1.5 km amsl) show
increases in concentrations for all variables under scrutiny, including $O_3$. The latter
(**Figure 5b**) is the most significant difference between the characteristics of features B
and A. Other differences include the much smaller increases in CO concentration
and OPC aerosol ($N_{PM1}$, $N_{PM2.5}$ and $N_{PM10}$) concentrations as well as extinction
coefficients observed in feature B (**Figure 5c, e, f**, respectively). The $O_3$/CO ratio is



also larger (i.e. 0.25, not shown) than that associate with feature A. These
observations, together with wind measurements, suggest that feature B corresponds
to a more aged urban plume. This could be an indication that the ATR 42 sampled
more than just the Lomé plume. This will be investigated using tracer experiments in
Section 6. Above 2 km amsl, the AAE increases to larger values (> 1.5), evidencing a
change in aerosol nature, i.e. a transition from local urban emissions to elevated
background pollution (**Figure 5g**), possibly resulting from a mixture long-lived
anthropogenic pollution and long-range transport of dust and biomass burning
aerosols from previous days.

Regarding feature C, the in situ measurements do not allow characterizing the
nature of the aerosols. The origin of this layer will also be investigated using tracer
experiments (see Section 6).

The in situ measurements along the elevated ATR 42 track reveal significant
differences in aerosol/gas phase concentrations and properties between the
western part (where feature E is observed with the lidar) and the eastern part (where
feature D is observed) of the ATR 42 leg (**Figure 6**). In the western part, ATR 42
measurements highlight enhanced $O_3$ and CO concentrations (> 60 ppbv and
> 200 ppvb, respectively, **Figure 6a, b**) together with AAE values of ~1.5 (**Figure 5f**),
suggesting the presence of biomass burning aerosol. Furthermore, aerosol number
concentrations $N_{PM1}$ and $N_{10}$ show enhanced values for small particles (100 # $cm^{-3}$
and ~1000 # $cm^{-3}$, respectively, **Figure 6c, d**). The observed $O_3$, CO and $N_{10}$
concentrations are larger than the background values measured during the ascent
over the ocean (~40 ppbv, 150 ppbv, and 500 # $cm^{-3}$, respectively, **Figure 5b, c, f**).





Large extinction values are also observed (100 Mm$^{-1}$), largely exceeding the
background value of 30 Mm$^{-1}$ (compare **Figure 6e and Figure 5e).**

In the eastern part of the leg, AAE values of ~1.5 also suggest that biomass burning
aerosols are sampled. O3, CO, $N_{PM1}$ and $N_{10}$ concentrations diminish approximately
half way through the leg to their background values (from 1716 UTC on, **Figure 6a, b,**
**c, d**), as does the extinction coefficient. However, $N_{PM2.5}$ and $N_{PM10}$ concentrations
increase significantly, as opposed to $N_{PM1}$, which combined with enhanced lidar-
derived VDR suggest mixing with larger particles, possibly dust. Further insight into the
origin of these aerosols, observed as a result of long-range transport, will be
investigated in Section 7.

Finally, in Section 8 we will investigate the cause of the slanting of the elevated
aerosol layers from west to east along the flight track, which also possibly leads, in
addition to the colder SSTs, to a thinning of the marine ABL and the suppression of
clouds at its top in the vicinity of Lomé (**Figure 3**).

6.   Tracer experiments for anthropogenic aerosols


The objectives of the tracer experiments are threefold: (i) understand how the lower
tropospheric circulation shapes the structure of the urban pollution plume emitted
from coastal cities and observed with the ULICE lidar (marked A and B in **Figure 3**), (ii)
assess which cities contribute to the plume observed with ULICE and whether it results
from Lomé emissions only, and (iii) provide insight into the origin of the intermediate
aerosol layer (marked B in **Figure 3**). For this we have analyzed along the ATR 42
aircraft flight track the tracer simulations introduced in Section 3.





As an ancillary objective, we also aim to assess how far over the ocean the urban
pollution aerosols can be transported by the complex low-level circulation over SWA.
For this, we have analyzed the tracer simulations along four 0.5°-wide north–south
transects spanning the longitudinal range of the ATR 42 flight (centered at 0.75°W,
0.25°W, 0.25°E and 0.75°E, cf. **Figure 1b**).

6.1 Structure of the urban plume along the coastline

**Figure 7** shows the structure of the urban pollution plume along the aircraft track
between 1400 and 1800 UTC in the TRA_D and TRA_I experiments. In TRA_D1 (**Figure**
**7a**), feature A as observed in the lidar VDR field (**Figure 3**) corresponds to emissions
from Lomé only (in greenish colors) in the ABL (blue dotted line), while feature B
corresponds to emissions from Lomé mainly with a contribution from Accra
(superimposed with the Lomé plume) and Cotonou (reddish colors in the upper
western boundary of the Lomé plume). In the TRA_I1 experiment, the Accra
contribution is missing altogether (**Figure 7b**). More strikingly, TRA_D1 shows an
elevated tracer plume over the ocean originating from Accra (blueish colors), which
mimics feature C in **Figure 3** fairly well. This feature is almost absent in TRA_I1, stressing
the importance of accounting for enhanced emissions from Accra (with respect to
Lomé and Cotonou) to produce a more realistic tracer simulation.

Results from experiment TRA_D2 (**Figure 7c**) shows that feature C in the lidar VDR
observations is likely related to emissions from Accra from the previous day only (i.e. 1
July), as the structure of the Accra plume in TRA_D1 and TRA_D2 is the same. In this
experiment, the structures of the plume corresponding to features A and B in **Figure 3**




are clearly altered by the lack of recent emissions in Lomé on 2 July (the lower part of
the plume is likely advected northward with the southerly flow here). This is confirmed
by looking at the result of TRA_D3 (**Figure 7d**): the fresh emissions (on 2 July) from
Lomé do lead to a realistic simulation of the shape of features A and B observed by
lidar. On the other hand, feature C is not reproduced in this experiment, suggesting
that feature B as observed by lidar is a mix of fresh and more aged emissions from
Lomé, as well as aged emissions from Cotonou and Accra, while feature C is almost
entirely related to aged pollution from Accra. What is also worth noting is that no
emissions from Lagos on 1 and 2 July are observed along the ATR 42 flight track in the
TRA_D and TRA_I experiments.

6.2 Southward transport of the urban plume over the Gulf of Guinea

**Figure 8** shows the structure of the urban pollution plume along four 0.5°-wide north–
south transects centered at 0.75°W, 0.25°W, 0.25°E and 0.75°E on 2 July at 1600 UTC,
i.e. half way through the ATR 42 flight.

Along the westernmost transect, labeled I in **Figure 1b** (centered at 0.75°W), the
pollution plume is only composed of emissions from Accra and is lifted off the surface
above the ABL (**Figure 8a**). Note that no tracer emissions directly occur in this
transect, with Accra emissions being contained in transect II, to the east of transect I.
As discussed by Knippertz et al. (2017), during the campaign, pollution plumes from
coastal cities were mostly directed northeastwards (see their Figure 19). Hence the
tracer plume seen in the experiment on 2 July is associated with transport of tracers
emitted on 1 July in the monsoon flow toward the northeast, which are then vertically
mixed (due to thermally and mechanically driven turbulence), and westward



advection of the tracers by the easterly flow above the monsoon layer. Over the
ocean, the plume is seen to extend as far south as 4.7°N, i.e. the southernmost
extension seen on all transects shown in **Figure 8**. This is linked to a small equatorward
component in the easterly flow (not a meridional overturning circulation).

Along the transect centered at 0.25°W (transect II, **Figure 1b**), the plume is seen to be
in contact with the surface as far north as 6.5°N (**Figure 8b**). The strong ascent at 6°N
is related to the presence of the Mampong range in the Ashanti uplands (see **Figure**
**1b**). The presence of the range and the associated motion contributes to deep
mixing of the plume north of Accra with the top of the tracer plume reaching 4 km
above the ground level or higher. Strong subsidence is seen north of the Mampong
range that mixes tracers down to the surface. Other ascending and subsiding
motions are detectable over the Lake Volta area, which could be related to land-
lake breeze systems. South of 6°N, the tracer plume is as deep as along transect I,
but does not extend southward over the ocean. Here also, only emissions from Lomé
contribute to the pollution plume on 2 July, suggesting that it took 24 h for these
emissions to reach transect II.

The pollution plume along the transect centered at 0.25°E (transect III) is structurally
similar to the one along transect II, but reaches farther inland (~7.5°N at the surface,
**Figure 8c**) than in transect II, likely due to the gap between the Mampong range
and the Akwapim-Togo range, and the flat terrain around Lake Volta. Again,
ascending and subsiding motions are detectable over the Lake Volta area that
could be related to land-lake breeze systems. Over the ocean, the plume reaches
5.3°N at 1.5 km amsl. Emissions from Lomé and Cotonou contribute to the upper and
southernmost part of the tracer plume along this transect, just north of 5.6°N.

648 Finally, along transect IV, the composition of the urban pollution plume is dominated

649 by emissions from Accra, with a small contribution of emissions from Cotonou and

650 Lomé in the southern, uppermost part of the plume because of short-range

651 westward transport above the monsoon flow (**Figure 8d**). The Accra plume is seen to

652 extend from the coastline to as far as 9°N and above the depth of the continental

653 ABL, but not as deep as along other transects with more pronounced orography. The

654 northward extension of the plume suggests that emissions from Accra are

655 transported over Togo along the eastern flank of the Akwapim-Togo range. Over the

656 ocean, the upper part of the plume barely reaches 5.6°N at an altitude of 2 km amsl.

658 The differences seen in the structure of the pollution plume obtained from the tracer

659 experiment over land are likely due to interactions between the monsoon flow and

660 the orography just to the north of Accra: namely the southeast–northwest running

661 Mampong range and the north-south running Akwapim-Togo range to the east of

662 Accra, both bordering Lake Volta (**Figure 1b**). In addition to those orographic effects,

663 the monsoon flow transporting the tracers towards the north may also interact with

664 the land-lake breeze system occurring in the summer over Lake Volta (Buchholz et

665 al., 2017a, b). Addressing the impact of these complex circulations over land on the

666 urban pollution plumes is beyond the scope of this paper.

668 Strikingly, as in the along aircraft flight track cross-section, emissions from Lagos on 1

669 and 2 July are never seen in the north-south transects, confirming that they likely do

670 not impact on the air quality in the major coastal cities to the west during this period.

671 Furthermore, the tracer simulations suggest that the pollution plume over SWA

672 related to emissions in the four cities considered here does not extend very far over



the ocean (to 4.7°N at most), essentially because they are transported northward
within and westward above the marine ABL. Nevertheless, the western part of the
Accra pollution plume spreads farther south over the ocean than the eastern part.

7.   Long-range transport of aerosols related to regional-scale dynamics


To gain insights into the origin of the aerosol layers sampled by the ATR along the
elevated leg and observed by lidar (features D and E in **Figure 3**), 10-day back-
trajectories ending at 2500 m amsl at 1700 UTC on 2 July are computed using
CHIMERE. The backplume associated with feature D is shown in **Figure 9a** (the one
associated with feature E is nearly identical and will not be discussed). The back
trajectories suggest that feature D originates from a broad area including Gabon,
Congo and the Democratic Republic of Congo. Most of the back trajectories then
travel over the Gulf of Guinea towards SWA in the free troposphere (**Figure 9b**). Daily
mean AOD derived from MODIS and SEVERI observations on 2 July (**Figure 10a**) show
large values offshore of Gabon and Congo known to be biomass burning aerosol
emission hotspots at this time of year (e.g. Menut et al., 2018). This is corroborated by
the CAMS biomass burning aerosol forecast at 1200 UTC (**Figure S4a**).

The afternoon CALIOP observations acquired to the east of the ATR 42 flight track
across the enhanced AOD feature (see track in **Figure 10a**) indeed classify the
aerosols over the ocean as elevated smoke, transported between 1.5 and 4 km amsl
(**Figure 10b**). The altitude of transport is consistent with that derived from the CHIMERE
backplume (**Figure 9b**) as also shown by Menut et al. (2018). Along this transect, dust
is observed to almost reach the SWA coastline from the north (**Figure 10b**) consistent
with the moderate AOD values observed over Togo and Benin (**Figure 10a**).





Furthermore, the morning ATR 42 flight conducted on 2 July in the region of Savè
(Benin, ~8°N) highlighted the presence of dust over northern Benin (Flamant et al.,
2018). Interestingly, at the coast (~6°N), CALIOP shows evidence of polluted dust,
possibly resulting from the mixing of dust with anthropogenic emissions from coastal
cities. However, the CAMS forecast does not show dust reaching the SWA coast
(**Figure S4b**).

The backplume and regional scale dynamics analyses indicate that the upper-level
aerosol features D and E (as observed by lidar) are related to biomass burning over
Central Africa. In the case of feature D, closer to Lomé, MODIS, SEVIRI and CALIOP
observations suggest the possibility of mixing with dust, consistently with the ATR in situ
and lidar-related observations.

8.  Coastal circulations: the role of surface temperature gradients and orography

IFS vertical velocity computed between 850 and 600 hPa (i.e. above the monsoon
flow) shows that most of the northern Gulf of Guinea is under the influence of
subsiding motion on 2 July at 1800 UTC (**Figure 11b**). Stronger subsidence is seen to
the east of the region of operation of the ATR 42 at that time. Strong subsidence is
also seen over the eastern part of the ATR 42 flight track at 1200 UTC (**Figure 11a**).
However, at 1200 UTC, the eastern part of the northern Gulf of Guinea is
characterized by upward motion, possibly in relationship with the SST gradient (cold
water to the west linked with the coastal upwelling and warmer waters to the east in
the Niger delta region). The signature of the sea breeze is also visible inland in the IFS
analysis at 1200 UTC (**Figure 11a**) in the form of a line of strong ascendance running
parallel to the coastline.




At the regional scale, IFS analyses evidence the existence of marked surface
temperature difference between the ocean and the continent at 1200 UTC (**Figure**
**S5d**) because of the high insolation across SWA as noted in Section 2. The surface
temperature gradient across the coast creates shallow overturning circulations as
evidenced by IFS analyses at 1800 UTC (**Figure 12**). A well-defined closed zonal cell
can be identified below 600 hPa around 5°N and between 0°E and 8°E (**Figure 12a**),
while a well-defined meridional cell is seen around 0°E between 3°N and 8°N (**Figure**
**12c**). It is worth noting that the overturning circulations are most intense and better
defined at 1800 UTC than at 1200 UTC (compare **Figure 12a** with **Figure S5c** for the
zonal cell), even though the surface temperature difference across the coast is
weaker (compare **Figure 12b** with **Figure S5d**). The overturning circulation exhibits a
strong diurnal cycle (**Figure S5**), which is driven by the surface temperatures over
land. The quality of IFS skin temperature during the day was verified against observed
land surface temperature observations (so-called Copernicus product; see **Figure**
**S6**). In spite of a systematic bias on the order of 2°C over land, IFS skin temperature
analyses are seen to be consistent (in terms of spatio-temporal distribution) with the
Copernicus product (**Figure S6**). This gives us confidence that the overturning
circulations exists and contributes to enhance subsidence over the Gulf of Guinea.
Furthermore, we have conducted an analysis of the correlation between the land-
sea skin temperature gradients associated with both the zonal and the meridian cells
and the vertical velocity over the Gulf of Guinea at different times of day for the
whole of July 2016, based on IFS data (**Table 2**). The analysis shows that the zonal
land-sea skin temperature gradient at 1200 and 1800 UTC is significantly correlated
with vertical velocity at 1800 UTC with values around 0.5. On the other hand, the
meridional land-sea skin temperature gradient at 1200 UTC is correlated (0.34) with



vertical velocity at 1200 UTC, possibly due to the presence of orography as discussed
in the following. Hence, the overturning cells evidenced on 2 July appear to be
persistent features over the Gulf of Guinea, at least in post-monsoon onset
conditions.

In addition to the subsidence generated at the regional scale by the land-sea
temperature gradient, the interaction of the monsoon flow with the orography over
Ghana and Togo is responsible for more local coastal circulations. This interaction is
reflected in the vertical velocity anomaly simulated with WRF along the western- and
easternmost transects in **Figure 1b** (transects I and IV, respectively). The anomalies
are computed with respect to the average vertical velocity between 1°W and 1°E.
**Figure 13** shows that in the region where orography is more pronounced (i.e. to the
west), the vertical velocity anomaly is positive, while it is negative to the east where
orography is less marked (compare **Figure 13a** and **13b**). As a result, the eastern
region of ATR 42 operation on 2 July is under the influence of strong subsiding motion.
This subsiding motion suppresses low-level cloudiness near Lomé and is key to the
interpretation of the ATR 42 lidar observations along the track regarding the slanting
of the elevated aerosol layers and, possibly, the thinning of the marine ABL towards
the eastern end of the aircraft track, together with an additional effect of colder
SSTs.

MODIS observations show the existence of an SST dipole across the northern part of
the Gulf of Guinea (**Figure S7** and **Figure 11**), between the coastal upwelling offshore
of Lomé and Accra (SSTs on the order of 26°C) and the warmer SST to the east in the
Bight of Bonny (offshore Nigeria, where SST on the order of 28°C are generally
observed). Even though this SST dipole may also generate a secondary circulation



over the Gulf of Guinea (e.g. around 900-800 hPa and between 0 and 1°E in **Figure**
**S5c**), it is very likely that the lower tropospheric dynamics in the region of operation of
the aircraft are dominated by the monsoon dynamics to the first order and by the
sea-land surface temperature gradient at the regional scale.

9.  Summary and conclusions

In this study, detailed aircraft observations and accompanying model simulations
were used to analyze the distribution of aerosols over the Gulf of Guinea and its
meteorological causes. We show that land-sea surface temperature gradients
between the northern part of the Gulf of Guinea and the continent as well as
orography over Ghana and Togo play important roles for the distribution of aerosols
and gases over coastal SWA. The former creates large-scale subsidence conditions
over the northern part of the Gulf of Guinea through the generation of zonal and
meridional overturning circulations below 600 hPa, with the downward branch of the
circulation around 0°E over the ocean. The latter generates enhanced subsidence
over the eastern part of the ATR 42 operation area, near Lomé and Accra. Together
this leads to a west–east tilting of the aerosol layers (that can be considered as
passive tracers of the dynamics) along the flight track. The ATR 42 sampled remotely
and in situ the complex aerosol layering occurring between 2.5 and 3.2 km amsl over
the Gulf of Guinea as a result of long-range transport of dust (from the northeast)
and biomass burning aerosol from the south (feature E in **Figure 3**) and the mixing
between these (feature D).

The orography-forced circulation also has an influence on the structure of the urban
pollution plumes from Accra, Lomé and Cotonou as assessed from airborne lidar
measurements and numerical passive tracer experiments using the WRF model.
When accounting for the relative size of the emitting cities along the coast (3 times
more emissions in Accra than in Lomé), we find that the tracer experiment designed
to include emissions from 1 and 2 July is the most realistic in reproducing the lidar
observations. The analysis shows that (a) the large pollution plumes observed at the
coast up to 1.5 km (features A and B) are essentially related to emissions in the Lomé
area from both 1 and 2 July, with a moderate contribution from Accra and Cotonou,
(b) the elevated plume over the northern part of the Gulf of Guinea (feature C) is
related to emissions from Accra exclusively from the day before the ATR 42 flight (i.e.
1 July) and these clearly dominate the composition of the tracer plume in the region
covered by the flight track on 2 July, (c) Lagos emissions (taken to be 20 times that of
Lomé) do not appear to be a player for regions west of Lomé in the summer in post-
onset conditions as also shown by Deroubaix et al. (2018), and (d) the tracer plumes
do not extend very far over the ocean, mostly because they are transported
northward within and westward above the marine ABL.

The unique combination of in situ and remote sensing observations acquired over
the Gulf of Guinea during the 2 July OLACTA flight together with global and regional
model simulations revealed in details the impact of the complex atmospheric
circulation at the coast on the aerosol composition and distribution over the northern
Gulf of Guinea. We show that the western Gulf of Benin is a place favorable for
subsidence in the afternoon due to 3 factors, namely cool SSTs, zonal overturning
connected with the Niger Delta region and meridional overturning connected with
the main West African landmass, anchored geographically at the Mampong and
Akwapim-Togo ranges. We also show that the overturning cells are robust features of
the atmospheric circulation over the Gulf of Guinea in July 2016. To the best of the



authors' knowledge such features have not been documented in the literature to
date.

Further research will be dedicated to enhance our understanding of the complex
interactions between the monsoon flow and the orography north of major coastal
cities as well as the land-sea and land-lake breezes, and their impact on the
dispersion of pollution emissions form major coastal cities in SWA. Future research will
also be conducted to assess long-term impact of the land-sea surface temperature
gradient (and related shallow overturning circulation) on distribution of aerosols over
the northern Gulf of Guinea.

**Acknowledgements**

The DACCIWA project has received funding from the European Union Seventh
Framework Programme (FP7/2007-2013) under grant agreement no. 603502. The
European Facility for Airborne Research (EUFAR, http://www.eufar.net/) also
supported the project through the funding of the Transnational Activity project
OLACTA. The Centre National d'Etudes Spatiales (CNES) provided financial support
for the operation of the ULICE lidar. The personnel of the Service des Avions Français
Instrumentés pour la Recherche en Environnement (SAFIRE, a joint entity of CNRS,
Météo-France and CNES and operator of the ATR 42) are thanked for their support.
M. Gaetani has been supported by the LABEX project funded by Agence Nationale
de la Recherche (French National Research Agency, grant ANR-10-LABX-18-01). The
authors would like to thank T. Bourrianne and B. Piguet (CNRM) and M. Ramonet
(LSCE) for their support in the data acquisition and processing. The authors would
also like to thank Gregor Pante (KIT) for providing IFS data, as well as Hugh Coe,



Sophie Haslett and Johnathan Taylor (Univ. of Manchester) for helpful discussions.
MODIS data was made available via the Geospatial Interactive Online Visualization
ANd aNalysis interface (https://giovanni.gsfc.nasa.gov/giovanni/).

**Data availability**

The aircraft and radiosonde data used here can be accessed using the DACCIWA
database at http://baobab.sedoo.fr/DACCIWA/. The tracer simulations discussed in
this paper are also available on the database. An embargo period of 2 years after
the upload applies. After that, external users can access the data in the same way
as DACCIWA participants before that time. Before the end of the embargo period,
external users can request the release of individual datasets. It is planned for
DACCIWA data to get DOIs, but this has not been realized for all datasets yet.


**Competing interests**

The authors declare that they have no conflict of interest.

**Special issue statement**

This article is part of the special issue "Results of the project 'Dynamics-aerosol-
chemistry-cloud interactions in West Africa' (DACCIWA) (ACP/AMT inter-journal SI)". It
is not associated with a conference.






**Appendix A: The ULICE lidar characteristics and data processing**

For the two channels of the lidar (indexed 1 and 2), the apparent backscatter coefficient (ABC, $\beta_{app}$) is given by

$$\beta_{app}^{1(2)}(r) = C^{1(2)} \cdot \left(\beta_m^{1(2)}(r) + \beta_a^{1(2)}(r)\right) \cdot \exp\left(-2 \cdot \int_0^r \alpha_a(r') \cdot dr'\right) \tag{A1}$$

where $\beta_m$ and $\beta_p$ are the backscatter coefficients for the molecular and the aerosol contributions, respectively; $\alpha_a$ is the aerosol extinction coefficient; $C^{1(2)}$ are the instrumental constants for each channel. The total ABC is given by:

$$\beta_{app}(r) = \frac{\beta_{app}^1(r) \cdot \left(1 + VDR(r)\right)}{C^1 \cdot \left(T_1^{//} + T_1^{\perp} \cdot VDR(r)\right)} \tag{A2}$$

where $T_i^{//}$ and $T_i^{\perp}$ are the transmissions of the co-polarization and cross-polarization contributions of the lidar polarized plate $i$, respectively. The VDR is thus given by the equation:

$$VDR(r) \approx \frac{T_1^{//} \cdot \beta_{app}^2(r)(r)}{R_c \cdot \beta_{app}^1(r)(r)} - \left(1 - T_1^{//}\right) \cdot \left(1 - T_2^{//}\right). \tag{A3}$$

The apparent scattering ratio (ASR, noted $R_{app}$) is expressed as:

$R_{app}(r) = \beta_{app}(r) / \beta_m^{//}(r). \tag{A4}$

As also shown by Chazette et al. (2012), the cross-calibration coefficient $R_c = C^2/C^1$ can be assessed by normalizing the lidar signals obtained in aerosol-free conditions, assuming the molecular VDR to be equal to 0.3945% at 355 nm, following Collis and Russel (1976). The dominant error source is the characterization of the plate transmission on the optical bench, which leads to a relative error close to 8% on the VDR (Chazette et al., 2012). During the DACCIWA field campaign, all lidar



measurements were conducted within aerosol layers and therefore we had to use
measurements performed just before the campaign during flight tests above the
Mediterranean Sea for assessing $R_c$.




Table A1. Summary of ULICE lidar characteristics

| ULICE lidar | Characteristics |
| --- | --- |
| **Emitter (Laser)** | Quantel Centurion, diode-pumped, air cooled |
|  | 6.5 mJ, 8 ns, 100 Hz @ 354.7 nm |
| **Laser divergence** | < 0.1 mrad |
| **Output beam** | Eyesafe ~40 × 30 mm beam, tunable 0 to 40 mrad divergence |
|  | with Altechna Motex expander (at 1/e²) |
| **Receiver** | 2 channels with the cross-polarisations |
| **Telescope** | Refractive,150 mm diameter, 280 mm effective focal length |
| **Field of view** | ~3 mrad |
| **Filtering** | Narrow band filters (200 pm) |
|  |  |
| **Detection** | Hamamatsu H10721 photo-multiplier tubes. |
| **Detection mode** | Analog |
| **Data acquisition** | 12 bits, 200 MHz sampling, 2 channels NI-5124 digitizer |
|  | manufactured by the National Instruments Company. |
| **Vertical sampling** |  |
| **Native** | 0.75 m |
| **After data processing** | 15-30 m |
| **Weight of the optical head** | ~20 kg |
| **Weight of the electronics** | ~10 kg |
| **Consumption** | 350 W at 24-28 V DC |










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



**Tables**

Table 1. SAFIRE ATR 42 payload. Only instruments used in this study are listed. The
complete payload is detailed in Flamant et al. (2018).

| Instrument | Parameter | Responsible institution |
|---|---|---|
| P (static & dynamic): Rosemount 120 & 1221 | Pressure | SAFIRE / CNRS |
| INS + GPS inertial units | Wind component, position | SAFIRE / CNRS |
| Adjustable (flow, orientation) Aerosol Community Inlet | Particle aerosol sampling D50 = 5 μm | CNRM / CNRS |
| Aircraft DUAL CPC counter MARIE | Particle number concentrations D>4 nm & D>15 nm (variable) 1 s time resolution | LaMP / UBP |
| OPC Grimm 1.109 | Ambient particle size distribution 0.25–25 μm 6 s time resolution | CNRM / CNRS |
| PSAP (3λ) | Absorption coefficient, black carbon content Blue 476 nm, green 530 nm, red 660 nm | LaMP / UPB |
| CAPS | Extinction $Mm^{-1}$ 530 nm 1 s time resolution | CNRM / CNRS |
| Mozart | CO, $O_3$ measured every second, then averaged over 30 s $O_3$: 1 ppbv; CO: 5 ppbv | SAFIRE / CNRS |
| TEI 42CTL NOx analyser | NOx; measured every 1 s, then | SAFIRE / CNRS |





| | averaged over n x 10s<br><br>50ppt integration over 120 s | |
|---|---|---|
| PICARRO | $CO_2$, $CH_4$, CO cavity ring down<br><br>spectroscopy<br><br>carbon dioxide ($CO_2$) every 5s with<br><br>precision 150 ppb; methane ($CH_4$) 1<br><br>ppb, and CO to a precision of 30 ppb | SAFIRE / CNRS |
| ULICE Aerosol / cloud lidar | Aerosol backscatter @ 355 nm | LSCE / UPMC |





Table 2. Correlation between vertical velocity and land-sea skin temperature
gradients at 0000, 0600, 1200 and 1800 UTC for July 2016. The land-sea zonal skin
temperature gradient is computed using a 'land box' defined as 6–9°E and 4.5–6.5°N
and a 'sea box' defined as 2–5°E and 4.5–6.5°N. The land-sea meridional skin
temperature gradient is computed using a 'land box' defined as 2°W–2°E and 6–8°N
and a 'sea box' defined as 2°W–2°E and 3–5°N. Vertical velocity is averaged in the
layer 850–600 hPa over a box defined as 2°W–2°E and 4–6°N. Correlations are
computed using vertical velocity and skin temperature gradient indices standardized
to 0000, 0600, 1200 and 1800 UTC means for the month of July 2016. Significant
correlations (and their p values) are given in bold.

| Zonal cell | | Vertical velocity | | | |
|---|---|---|---|---|---|
| | | 0000 UTC | 0600 UTC | 1200 UTC | 1800 UTC |
| Skin temperature gradient | 0000 UTC | 0.26 | -0.04 | 0.12 | -0.17 |
| | 0600 UTC | | -0.08 | 0.09 | 0.11 |
| | 1200 UTC | | | 0.02 | **0.53 (p=0.002)** |
| | 1800 UTC | | | | **0.46 (p=0.01)** |
| Meridional cell | | Vertical velocity | | | |
| | | 0000 UTC | 0600 UTC | 1200 UTC | 1800 UTC |
| Skin temperature gradient | 0000 UTC | 0.07 | -0.22 | 0.06 | -0.07 |
| | 0600 UTC | | -0.01 | 0.01 | -0.06 |
| | 1200 UTC | | | **0.34 (p=0.06)** | -0.24 |





| | 1800 UTC | | | | 0.20 |
|---|---|---|---|---|---|





**Figures**

(a)

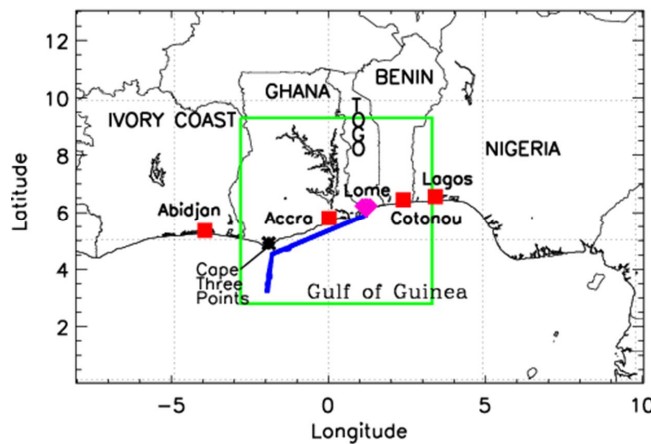

(b)



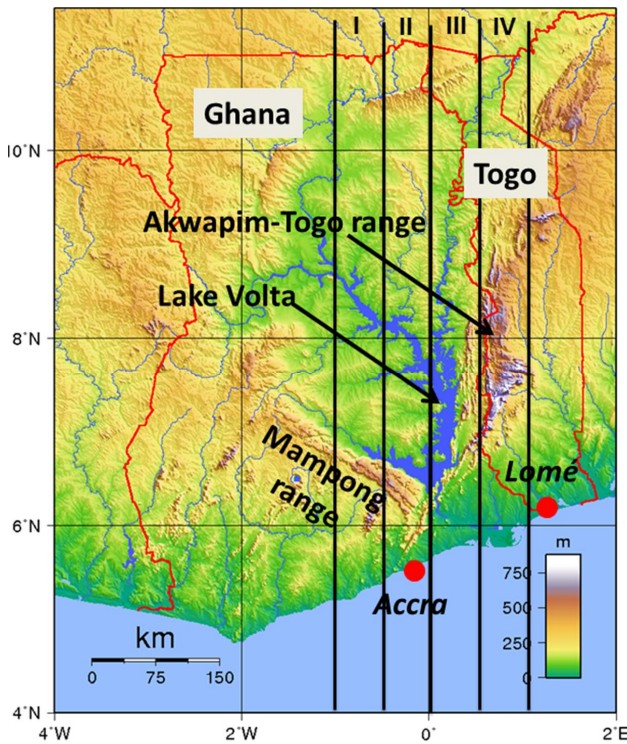

Figure 1: (a) Map of southern West Africa with the location of the main landmarks (e.g. cities, countries). The thick blue line represents the ATR 42 flight track in the afternoon of 2 July 2016. The red filled square symbols represent DACCIWA radiosounding stations used in this study. The pink filled circle represents the base of operation for aircraft during the DACCIWA field campaign. The green thick box represents the domain of the 2-km WRF simulation. (b) Topographic map of Ghana and Togo showing the main features of interest for this study as well as the transects along which tracer simulations are shown in **Figure 8**. The transects are centered at 0.75°W, 0.25°W, 0.25°E and 0.75°E (for I, II, III and IV, respectively) and are 0.5° wide.





(a)

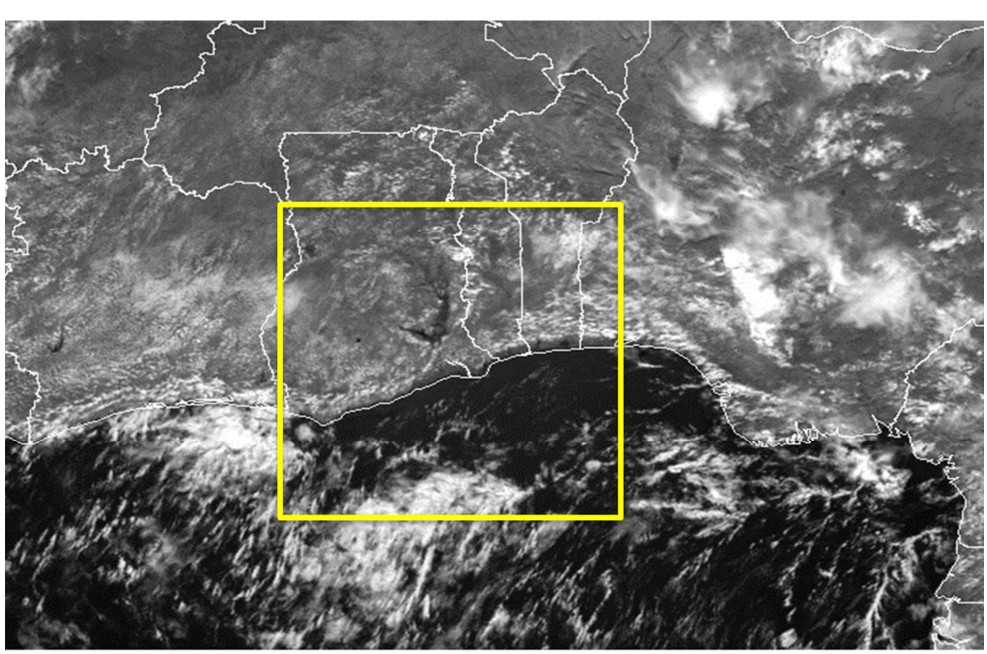

(b)

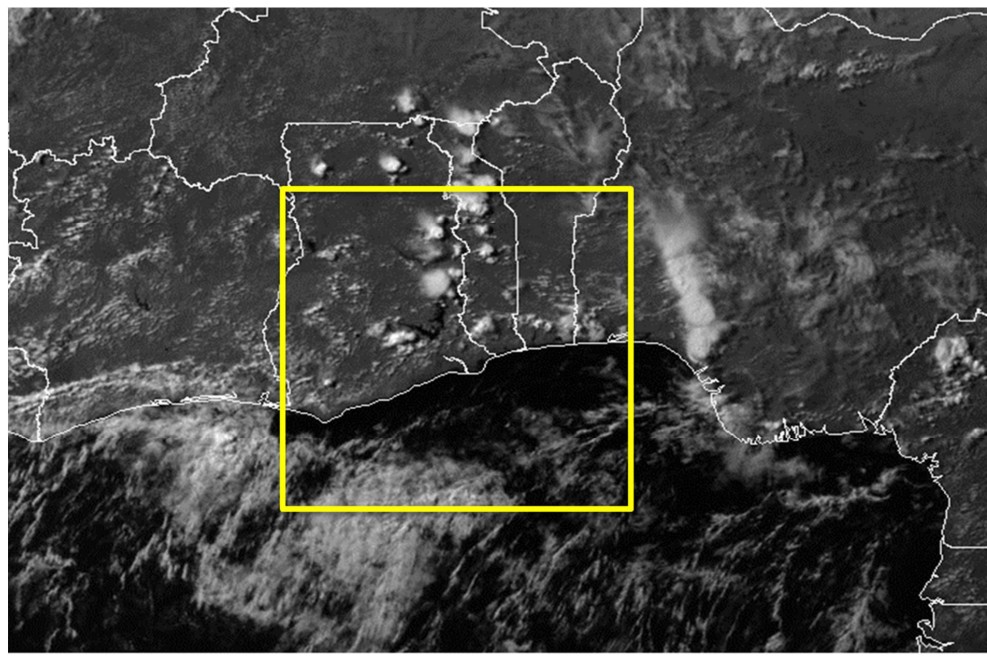





Figure 2: SEVIRI visible images of SWA on 2 July at (a) 1200 UTC and (b) 1500 UTC.
Country borders are shown as solid white lines. The yellow thick box represents the
domain of the 2-km WRF simulation as in **Figure 1a**.




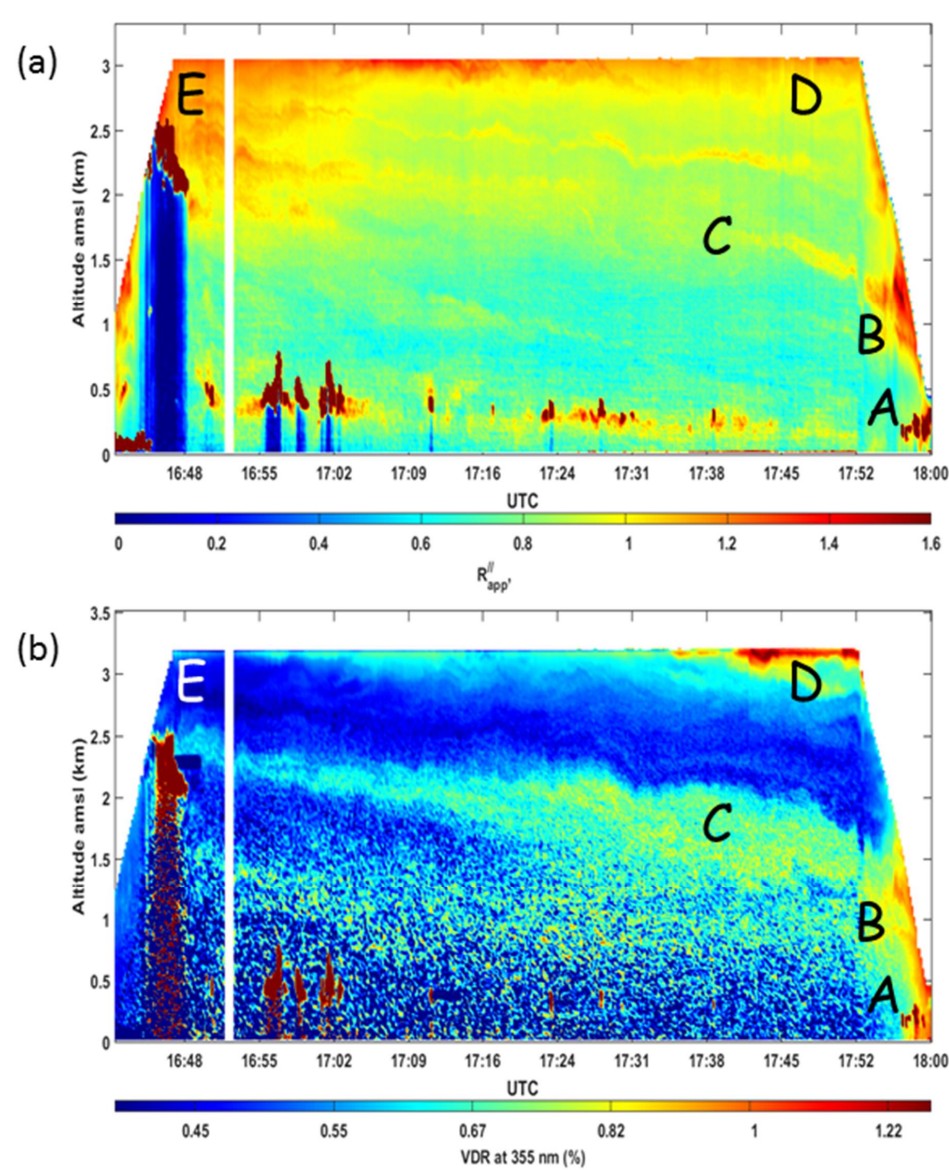


Figure 3: Time-height evolution of ULICE-derived (a) apparent scattering ratio ($R_{app}$)

and (b) volume depolarization ratio (VDR) below the ATR 42 flight track over the Gulf

of Guinea between 1644 and 1800 UTC on 2 July 2016 (see **Figure 1a**). The ATR leg





parallel to the coastline starts at 1654 UTC. The ATR passed the longitude of Accra at
1729 UTC. See text for explanations of features A–E.





(a)                          (b)

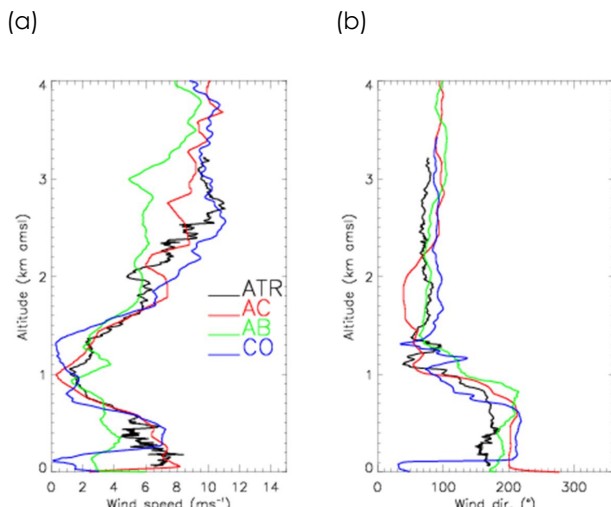


Figure 4: (a) Wind speed and (b) wind direction profiles measured during the ATR 42
sounding over the ocean (1630 to 1647 UTC, ATR, black solid line) as well as from the
radiosoundings launched in Accra at 1700 UTC (AC, red solid line), in Abidjan at 1608
UTC (AB, green solid line) and in Cotonou at 1612 UTC (CO, blue solid line). The
location of the radiosounding sites is shown in **Figure 1a**.




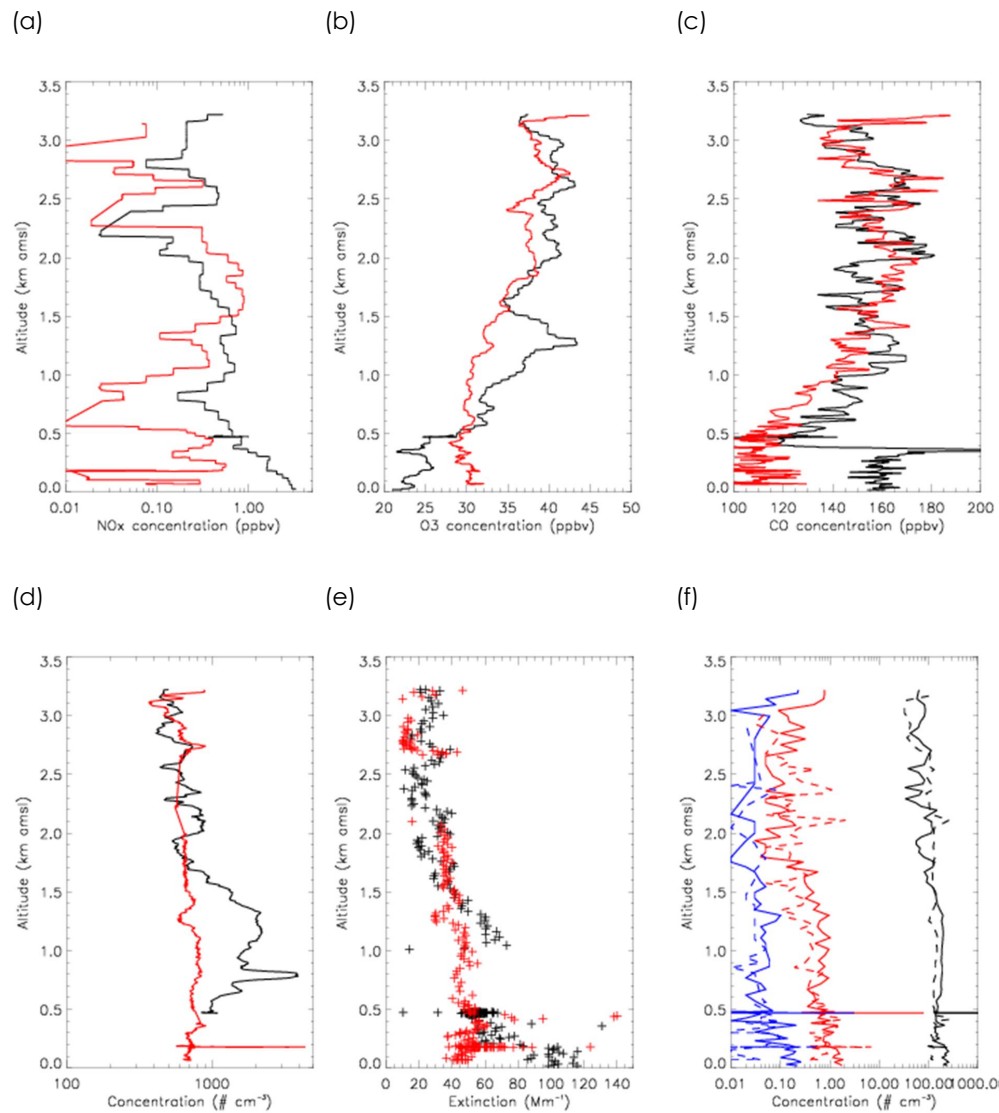



(g)

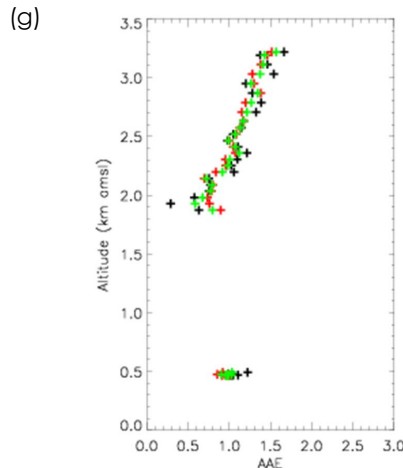

Figure 5: Profiles measured during the ATR 42 sounding over the ocean (1633 to 1647 UTC, red solid line) and at the coast in the vicinity of Lomé (1753 to 1807 UTC, black solid line) for (a) $NO_x$ concentration, (b) $O_3$ concentration, (c) CO concentration, (d) total aerosol concentration $N_{10}$ measured with the CPC and (e) extinction coefficient. (f) $N_{PM1}$, $N_{PM2.5}$ and $N_{PM10}$ concentration profiles (black, red and blue, respectively) measured over the ocean (dashed lines) and at the coast in the vicinity of Lomé (solid lines). (g) AAE profiles in the vicinity of Lomé computed between 467 and 530 nm, 530 and 660 nm, and 467 and 660 nm (black, red and green solid symbols, respectively).



1192

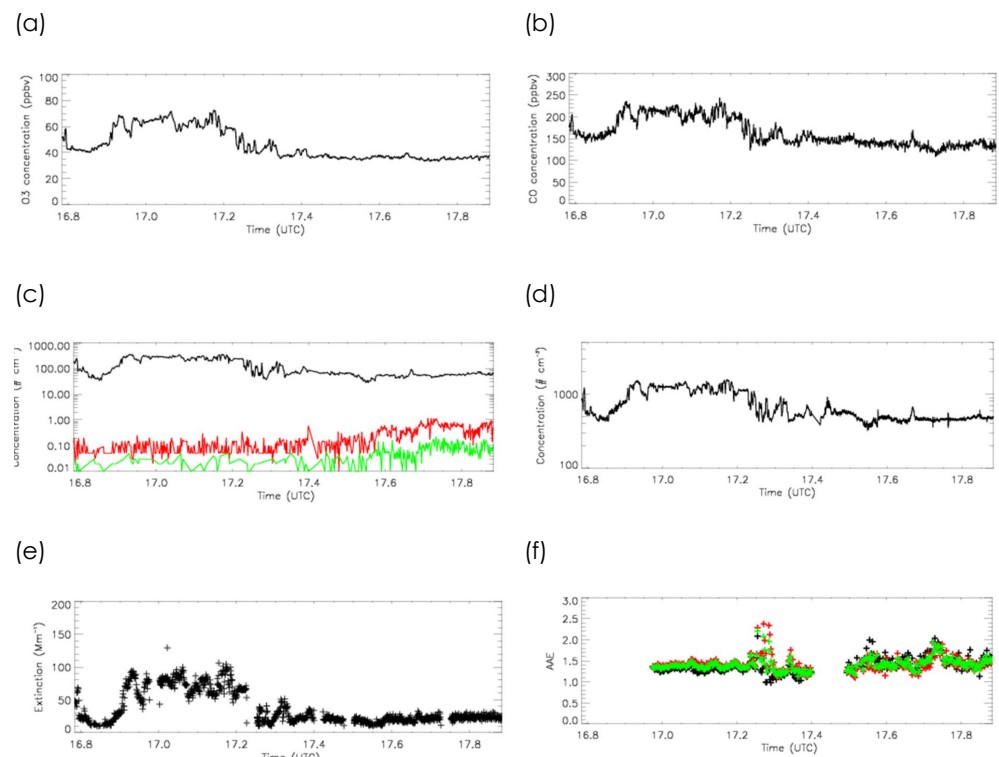

Figure 6: (a) $O_3$ concentration, (b) CO concentration, (c) $N_{PM1}$, $N_{PM2.5}$ and $N_{PM10}$ concentrations (black, red and green, respectively), (d) CPC-derived total aerosol concentration $N_{10}$, (e) extinction coefficient and (f) AAE computed between 476 and 530 nm, 530 and 660 nm, and 476 and 660 nm (black, red and green crosses, respectively) measured during the ATR 42 elevated straight level run from 1647 to 1753 UTC.







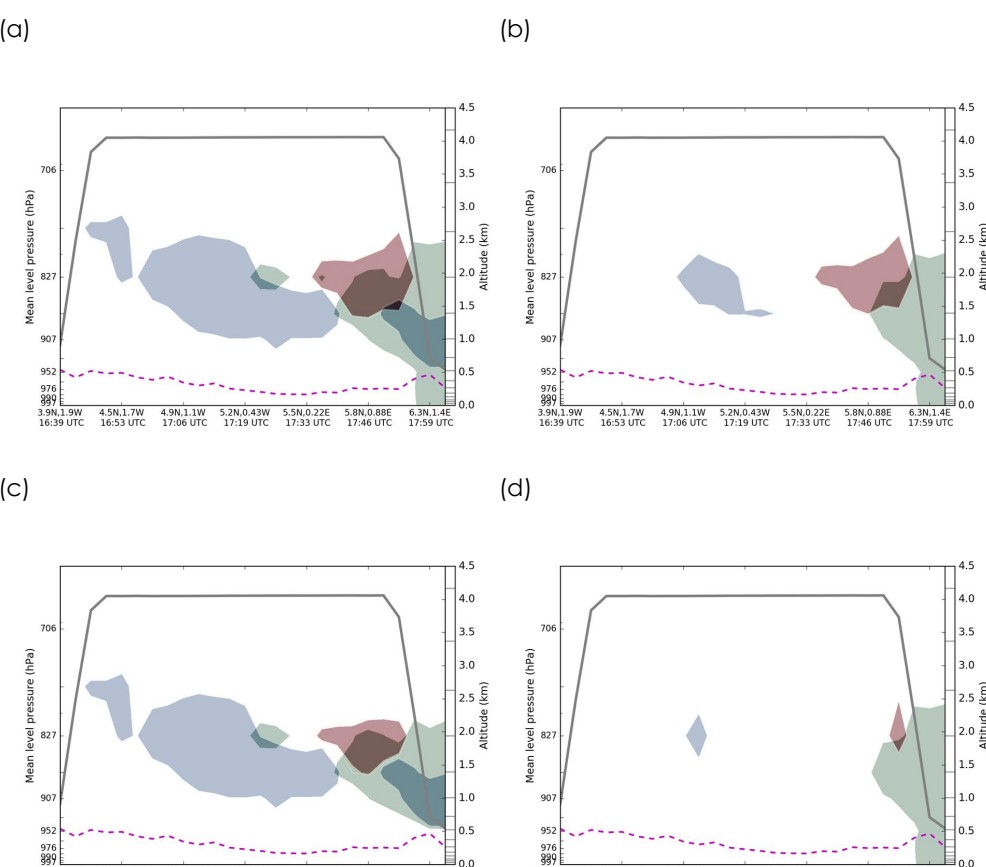

Figure 7: Time-height evolution of tracer concentration (a.u.) below the ATR 42
between 1400 and 1800 UTC for (a) the TRA_D1, (b) TRA_I1, (c) TRA_D2 and (d)
TRA_D3 experiments (see section 3.2.1 for details). Tracer emissions in Accra, Lomé
and Cotonou appear in blueish, greenish and reddish colors, respectively. The solid
grey line represents the altitude of the aircraft. The dashed blue line represents the
height of the top of the marine ABL from the WRF 2-km simulation.




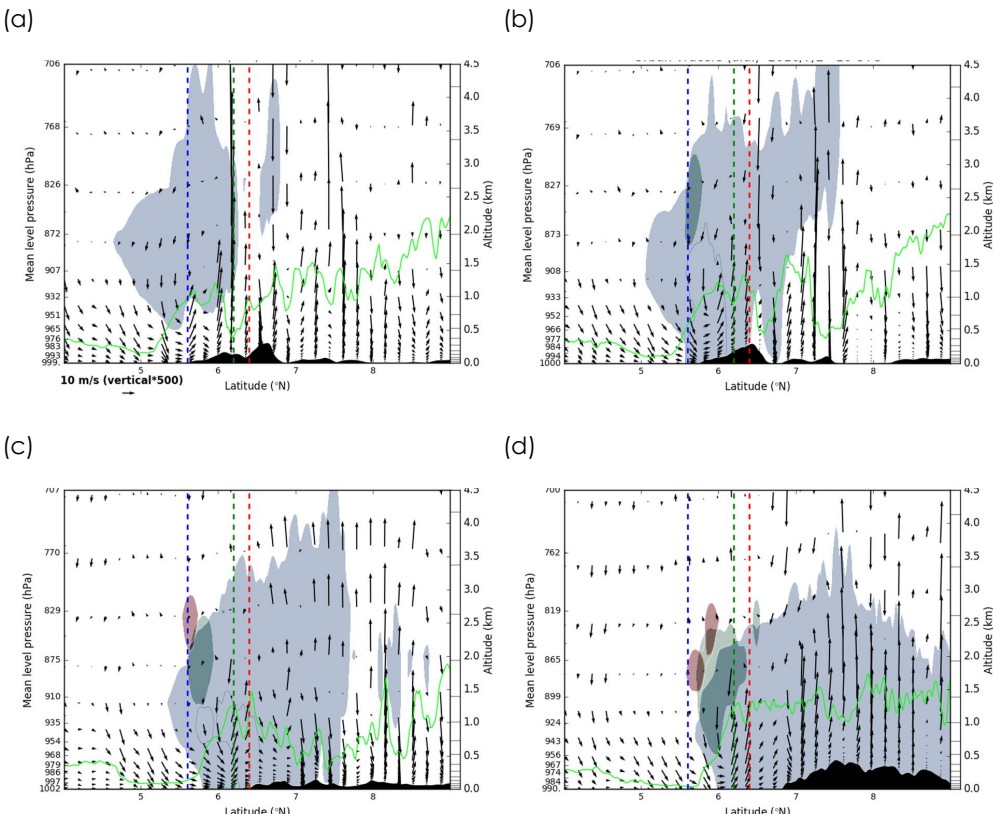

Figure 8: Tracer concentrations (a.u.) from the TRA_D1 experiment (see section 3.2.1
for details) along four 0.5°-wide north–south transects centered on (a) 0.75°W, (b)
0.25°W, (c) 0.25°E and (d) 0.75°E (marked I, II, III and IV, respectively, in **Figure 1b**) at
1600 UTC. Tracer emissions in Accra, Lomé and Cotonou appear in blueish, greenish
and reddish colors, respectively, as in **Figure 7**. Also shown are meridional-vertical
wind vectors in the transects. The green solid line represents the ABL derived from the
WRF 2-km simulation. The vertical dashed lines represent the location of the cities of
Accra (blue), Lomé (green) and Cotonou (red). The orography along the transects is
shaded in black.

(a)

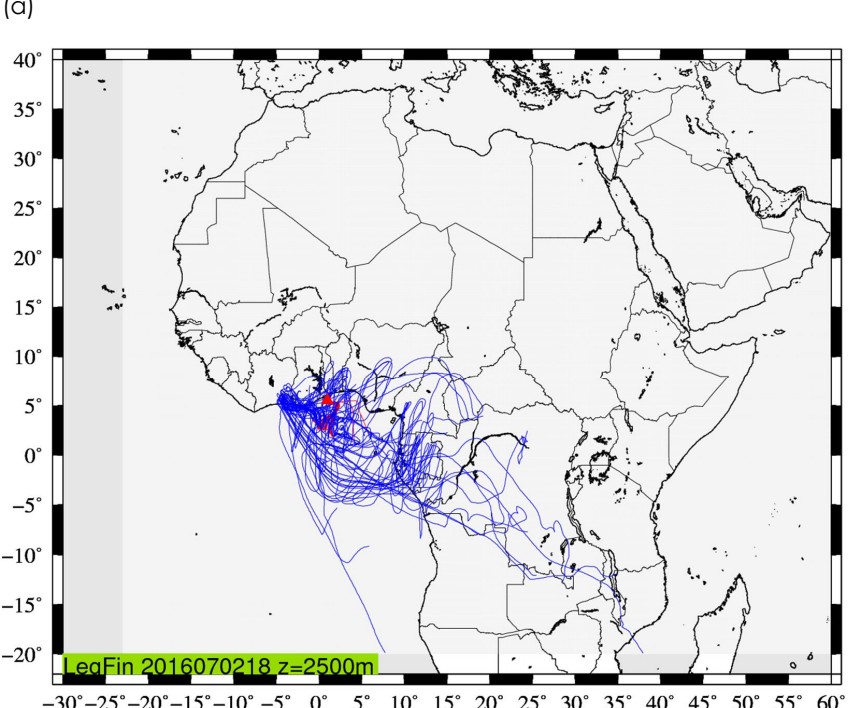

(b)

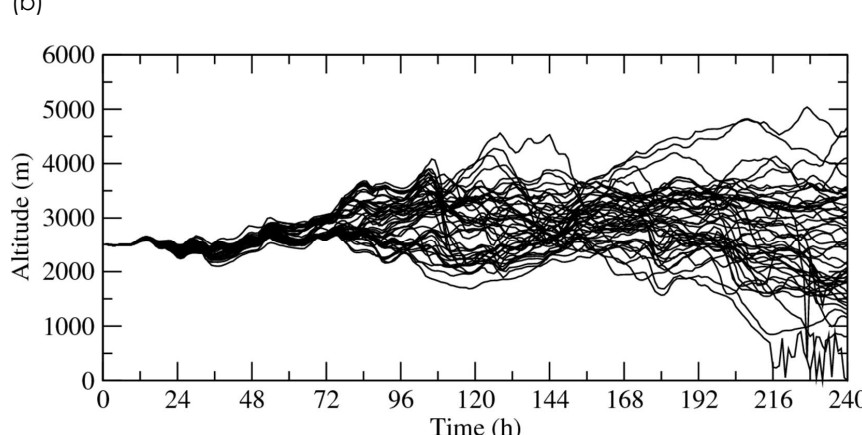


Figure 9: 10-day CHIMERE-derived backplume ending at 2500 m amsl at 5.5°N/1°E at

1700 UTC on 2 July 2016. (a) Individual trajectories are shown as blue solid lines over

a political map of Africa with state borders appearing in black. The red triangle



indicates the location of the origin of the back trajectories. (b) Time-height
representation of the individual back trajectories shown in the top panel.



(a)

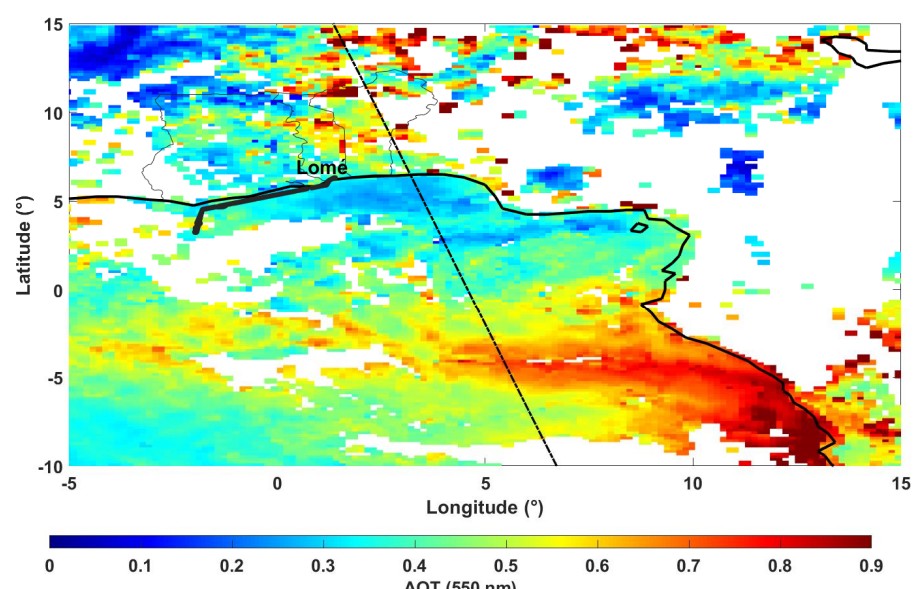

(b)

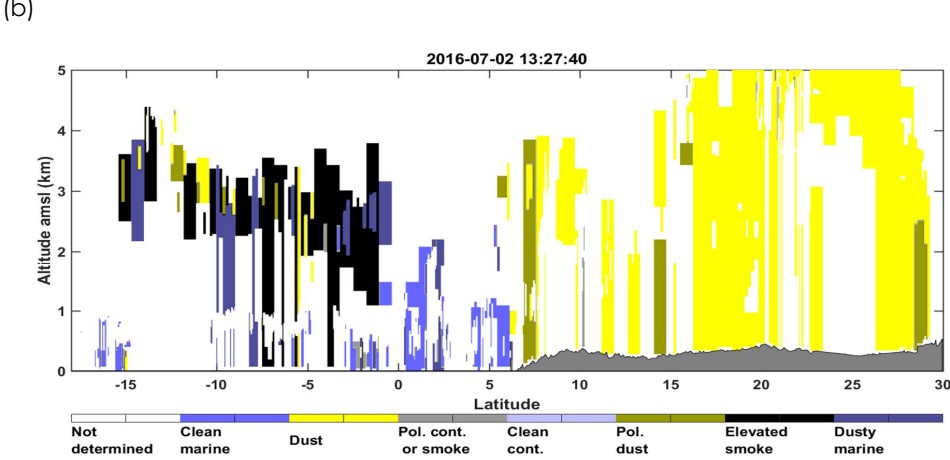

Figure 10: (a) Daily AOD obtained by averaging MODIS Dark target AOD (at 1325

UTC) and SEVIRI AOD (daily mean) on 2 July 2016. White areas indicate missing data.

Country borders of Ghana, Togo and Benin are shown as thin solid black lines. The

straight dashed-dotted line indicates the location of the CALIOP afternoon overpass



at 1327 UTC. The thick solid black line represents the ATR 42 flight track. (b) CALIOP-
derived aerosol classification for the afternoon overpass.





(a)

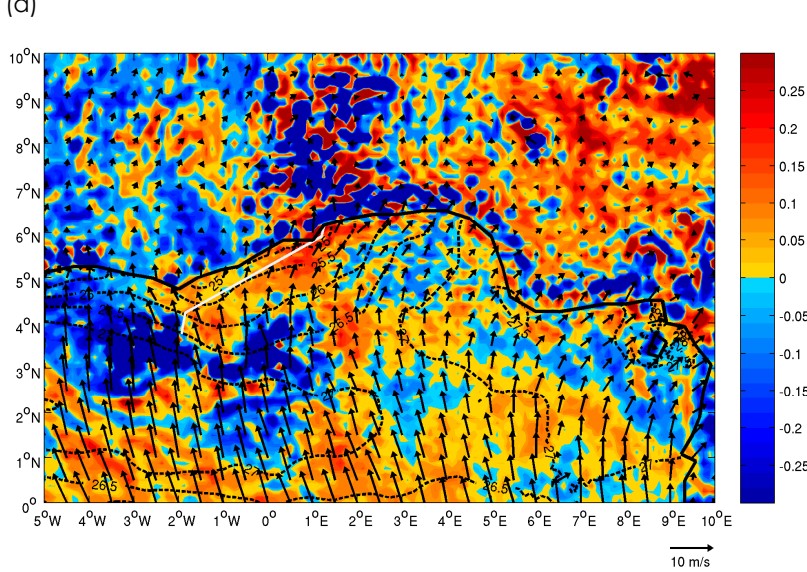

(b)

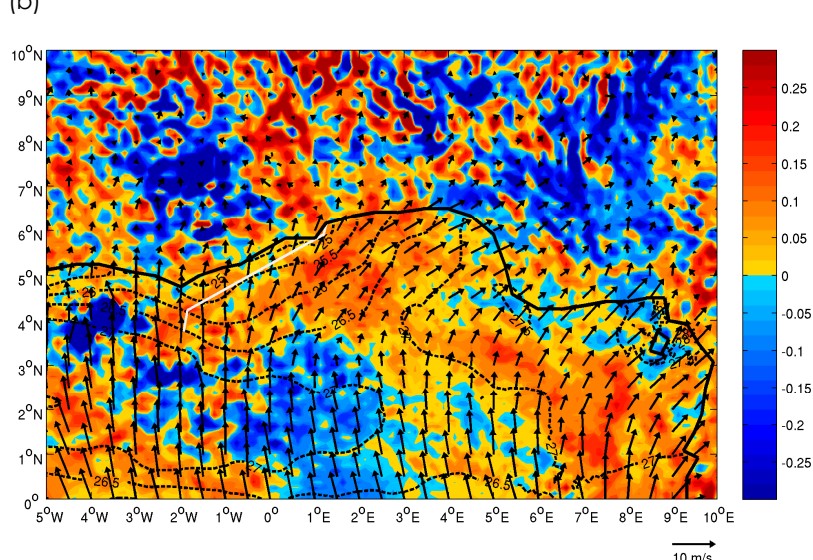


Figure 11: Vertical velocity averaged between 850 and 600 hPa (color, Pa s⁻¹) with

10-m winds (vectors) and SST (contours, black dotted lines) from IFS analyses at (a)



1200 UTC and (b) 1800 UTC. The thick black line represents the SWA coastline. The
straight white line represents the ATR 42 flight track.




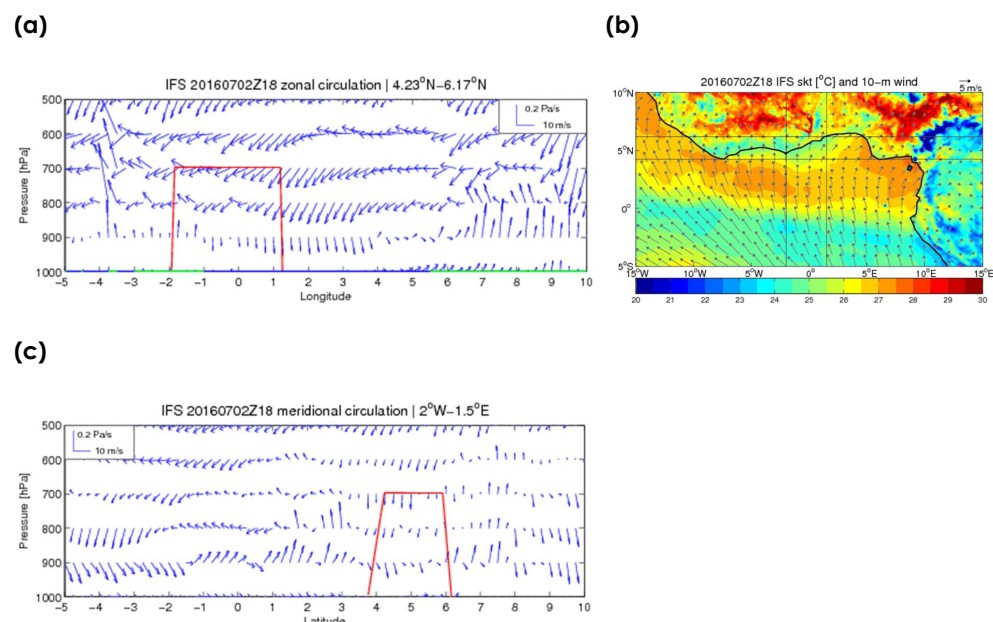

Figure 12: (a) West–east oriented vertical cross section (1000–500 hPa) of zonal-
vertical wind vectors from IFS analyses (blue) between 5°W and 10°E averaged
between 4.54°N and 6.17°N at 1800 UTC on 2 July 2016. The thick red line is the
projection of the ATR 42 aircraft track onto the cross-section. The thick green and
blue lines at the bottom of the graph indicate the presence of land and ocean,
respectively. Surface characteristics are defined based on the dominating surface
type in the latitudinal band considered for the average of the wind field. (b) IFS skin
temperature (colors) and wind field at 10 m (vectors) at 1800 UTC. The former,
originally at 0.125° resolution, has been linearly interpolated onto the Copernicus grid
at 5 km before computing the skin temperature differences between the
observations and the model. (c) North-south oriented vertical cross section (1000–500
hPa) of meridional-vertical wind vectors from IFS analyses (blue) between 5°S and
10°N averaged between 2°W and 1.5°E at 1800 UTC. The thick red line is the





projection of the ATR 42 aircraft track onto the cross-section. Cross-sections shown in
(a) and (c) are computed in the zonal and meridian windows delimited east-west
and north-south lines, respectively, shown in (b).






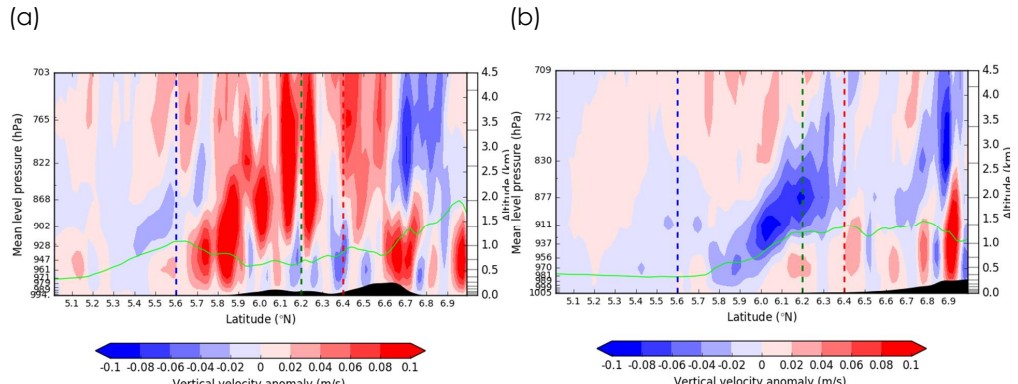


Figure 13: Vertical velocity anomaly along (a) the western most transect shown in
Figure 1b (transect I) and (b) the eastern most transect shown in Figure 1b (transect
IV), from the WRF 2-km simulation. The anomalies are computed with respect to the
average vertical velocity between 1°W and 1°E.