# Peer review of "Aerosol distribution in the northern Gulf of Guinea: local anthropogenic"

_Atmospheric Chemistry and Physics, 2018_

## Referee Comment (RC1) · Anonymous Referee #1 · 13 Jun 2018

**General Comments**

This paper analyzes aircraft observations of aerosols collected along the coast of South West Africa during the DACCIWA field campaign in June-July 2016. The authors go on to speciate the observed aerosol types and identify the likely aerosol emission sources and atmospheric dynamics that led to their transport and eventual spatial distribution recorded during the case study. The paper is well-written, well within the scope of ACP, and it is refreshing to see an observational study from this region, which has historically been observationally-sparse, making it a new addition to the scientific literature. Overall, it is easy to follow the narrative and methodology of the paper, although there are a few places that may need clarification or further explanation, which are mentioned below.

It would be nice to see a description on why this day of the field campaign was chosen for analysis. It seems like there is two months' worth of data from this project, so what makes 02-July-2016 so unique that it warrants its own paper, and how representative is it of typical flow patterns for this regime in the region? The primary concerns raised in the specific comments section regard aerosol aging and water uptake in humid environments, and timing of tracer release and the interpretation of maximum aerosol extent in the model.

It is recommended that the manuscript be published in ACP after the specific and technical comments are addressed in the paper.

**Specific Comments**

Page 2 - Lines 32-34) States that the lower troposphere aerosol loading includes emissions from Lagos, but later in the paper the tracer experiment shows that the aerosol plumes over the ocean do not have a signal from Lagos. Is this a reference to Lagos being an aerosol source in the SWA region, instead of the over the limited ocean aircraft data from this case study?

Page 4 - Lines 81-84) Is the purpose of DACCIWA / this paper to understand how atmospheric dynamics influences aerosol emission rates (e.g. stronger surface winds will loft more dust), or only aerosol transport after emission, or both?

Page 7 - Lines 161-163) Even though the optical properties could not be retrieved with the ULICE lidar inversion procedures, they were retrieved using other instrumentation, correct? If not, what was excluded?

Pages 7-8 - Lines 178-186) Can sea salt be identified with this method?

Pages 7-8 - Lines 178-186) What happens when there is a mixture of aerosol species instead of homogeneous plumes? Looking at Figure 10-b, the CALIOP data suggests a heterogeneous aerosol air mass during this case study event (e.g. dust mixing with smoke).

Pages 7-8 - Lines 178-186) Because the aircraft measurements were taken over the ocean, the particles reside in a relatively humid atmosphere. Depending on the aerosol species and the humidity of the environment, particles can take up water, changing their diameter and their optical properties. Does this affect VDR values or any metric by which the aerosol species were partitioned? Would it change the analysis in later sections at all, especially pertaining to attribution of fresh versus aged plumes?

Page 8 – Lines 181-182) What about urban O3? Will that mislead the speciation between smoke and pollution?

Page 8 – Line 202) It is stated that "data were processed with a time resolution of 1 s" – is this for all data or just the CAPS-Mex data? Was there some standard time resolution used for interpolation across instrumentations to line up the time resolutions? If so, what interpolation technique was used?

Page 13 – Lines 322-324) Is this one-way or two-way nesting in WRF?

Page 13 – Lines 326-327) More description of the WRF setup and physics options is necessary, especially the PBL parameterization, since the WRF PBL height is used later on in the paper. Furthermore, the WRF parameterizations used generally get a reference citation. Does the statement that the model configuration is the same as in Deroubaix et al. 2018 mean that every physics option / parameterization is identical to their setup? What about time steps, output intervals, and nudging? The Deroubaix et al., 2018 simulation was for a similar region in SWA, but the grid spacing was coarser, the simulation was run for a much longer duration to study short-term climate phenomena, and they ran with active chemistry instead of tracers. Stating that the setup is the same as in Deroubaix et al. 2018 may be confusing when these differences are considered.

Page 14 – Lines 344-346) Is there any observational evidence or prior literature that supports scaling urban emissions by population in this way? For example, why couldn't an efficient metropolis have 5x the population as a baseline city, but only 2x the pollution? Does the linear scaling of population and pollution break down at some point for this region or other regions?

Page 14 – Lines 347-349) The naming of the simulations is a bit counterintuitive. Instinctively, I'd think that TRA_D1 would represent July 1$^{st}$ and TRA_D2 as July 2$^{nd}$. However, TRA_D2 is July 1st and TRA_D3 is July 2$^{nd}$. By the time these simulations were discussed 13 pages later, the numbering became confusing. Perhaps numbering related to the dates would help readers later on (e.g. TRA_D12 = July 1$^{st}$-2$^{nd}$, TRA_D1= July 1$^{st}$ only, TRA_D2 = July 2$^{nd}$ only).

Page 14 – Lines 353-355) What does it mean that the lifetime of the tracers is designed to be 48 hours? Why set the concentration to zero if they are still present in the domain after 48 hours? Is it because the tracers do not undergo gravitational settling? Would including the gravitational settling process change the interpretation in later sections?

Page 15 – Line 365) Why are the tracers released at 2500 m ASL?

Page 18 – Lines 462-465) Maybe the placement of the 'A' on Figure 3 is misleading. To me, it looks like the 'A' is pointing to shallow clouds and not an aerosol layer.

Page 20 – Line 503) Is there an explanation for why there is a reduction in O3 concentrations compared to background levels for Plume A?

Page 20 – Line 506) What is the significance of the O3 to CO ratio? Why does the value of 0.15 imply the plume is fresh versus a value of 0.25 implies that it is aged?

Page 23 – Lines 592-593) This is regarding the statement that the emissions come only from July 1$^{st}$. Figure 4 shows the wind speeds above 500 m to be weak (1-2 m/s), so the emissions on July 2$^{nd}$ haven't had a chance to be advected far from their source regions in the weak winds. It makes sense then that

the emissions must be from July 1$^{st}$, or an earlier date. Is it possible that due to the low wind speeds above the PBL that what we are seeing isn't just from July 1$^{st}$, but also June 30$^{th}$? Would the picture change if the tracers were released starting on June 30$^{th}$?

Page 26 – Lines 671-675) Do you think the maximum extent that the plume reaches over the ocean in the model is related to the tracer lifetime and the end time of the simulation? If the simulation was run for longer, would the maximum tracer extent over ocean increase? This goes back to the previous comment about releasing tracers on June 30$^{th}$. If the tracers have no settling velocity or cannot be scavenged by precipitation, they could be advected indefinitely in the model.

Page 30 – Lines 751-752) Why is the correlation here related to terrain? I'm not sure I see the connection between skin temperature, vertical velocity, and terrain.

Page 35 – Lines 897-905) Was the flight over the Mediterranean an aerosol-free environment for calibration? If not, how might that affect the accuracy or uncertainty in the retrievals?

Page 47 – Table 1) Not every entry has a time resolution associated with it. Also, if uncertainty estimates are available they should be listed here.

Page 65 – Figures a,b) From CALIOP we have aerosol speciation, as well as horizontal and vertical location, and from MODIS we have some idea of the concentration. What new information did the aircraft observations and tracer experiments provide the community that we did not already have with the MODIS AOD and CALIOP data?

Page 2 Sup. Mat. – Lines 31-32) What is meant by variability across WRF grid boxes? Is this a standard deviation?

Page 2 Sup. Mat.) Was there moisture information available from the radiosondes or flight instruments? If so, how well did WRF do compared to the observations in terms of moisture? This also goes back to the point raised for Pg. 7-8 on how humid the environment was for this case study and how that might affect the retrievals.

Page 3 Sup. Mat.) Is this following the trajectory of the balloon and matching it to the WRF grid boxes, or assuming it is constant in horizontal model space at the release site lat/lon at the surface?

**Technical Comments**

Page 18 – Line 452) The word 'Possibly' should be lowercase

Page 18 – Line 457) Should this be Figure 5c and 5e instead of 4c and 4e?

Page 21 – Line 524) Missing word 'of' between 'mixture long-lived'

Page 23 – Line 581) Magenta line

Page 32 – Line 803) The WRF / CHIMERE models

Page 35 – Line 886) Subscript 'a' on beta instead of 'p'

Page 61 – Line 1205) Magenta line

Page 69 – Figure 12-c) Green and blue reference lines for land / ocean missing

Page 3 Sup. Mat. – Line 38) Missing UTC from 1700 and 1612

Page 4 Sup. Mat.) Missing a reference arrow for wind speeds

---

## Referee Comment (RC2) · Anonymous Referee #2 · 15 Jun 2018

Synopsis: This paper presents a day in the life of the airborne Dynamics-Aerosol-Chemistry-Cloud Interactions in West Africa project. Conclusions are drawn regarding the role of both synoptic and mesoscale meteorological features as well as the contributions nature of different sources on the aerosol environment. Overall, it is a reasonable analysis, but given it is really a one day analysis, it is difficult to support their findings in general. I myself use "a day in the life" sorts of papers to describe various phenomenon in a region in detail. But, such papers are always in a context of subsequent papers that then generalize. Here, the single day is used to generalize, which almost

by definition leads to unsupportable overall conclusions. e.g., can you really say a city's emissions are unimportant at some point based on a single day's analysis? This particular flight was pretty much parallel to the coast, such that the real littoral transition was never observed. I strongly recommend that the paper be reworked such that this one demonstrates key features. Showing a day in the life of the role if individual cities or meteorological phenomenon is worthy of publication if framed that way. But, generalization will need to happen with the support of a much more comprehensive airborne, satellite and modeling study of the entire field campaign to determine. As is I am not sure what to make of the paper or how it will be used by the community. Most of the work here is wordsmithing, so I do not think it is an overly onerous task to reframe in this way.

I pretty much agree with the other reviewer on specifics, where again a lot of generalization is made on a single observation. Here are a few more minor comments to consider.

On using AAE to speciate-line118: I am a bit concerned about using the AAE to say what the makeup of particles are given that by the analysis here there is often a mixture of aerosol species. This is further complicated for dust, which from aircraft inlets have a low penetration efficiency.

CAPS and Nephelometer-line 203: Again, the authors need to be mindful of dust particle penetration efficiencies and what that does to the interpretation of their results? I bring this up because based on the sounding of of figure 4 this level is in an area of some wind shear.

Figure 3 and 6. Instead of using time as an x axis, can you please use distance or perhaps longitude (given the aircraft track) so we can get a spatial perspective.

Figure 5-F. As well as number, can you please provide a profile of aerosol volume? It is much easier to interpret.

Figure 9. What happens if you have a minor change in altitude of release? This will show you how sensitive your system is.

---

## Author Comment (AC1) · 14 Jul 2018

**Aerosol distribution in the northern Gulf of Guinea: local anthropogenic sources, long-range transport and the role of coastal shallow circulations**

**By C. Flamant et al.**

Reply to the referees' comments

In the following, the comments made by the referees appear in black, while our replies are in red, and the proposed modified text in the typescript is in blue.

**Referee #1 comments**

**General Comments**

This paper analyzes aircraft observations of aerosols collected along the coast of South West Africa during the DACCIWA field campaign in June-July 2016. The authors go on to speciate the observed aerosol types and identify the likely aerosol emission sources and atmospheric dynamics that led to their transport and eventual spatial distribution recorded during the case study. The paper is well-written, well within the scope of ACP, and it is refreshing to see an observational study from this region, which has historically been observationally-sparse, making it a new addition to the scientific literature. Overall, it is easy to follow the narrative and methodology of the paper, although there are a few places that may need clarification or further explanation, which are mentioned below.

We would like to thank the reviewer for his/her mindful and benevolent comments on the paper. We have worked hard to comply with all of them. We now also acknowledge the work of the anonymous referee in the acknowledgement section of the paper.

It would be nice to see a description on why this day of the field campaign was chosen for analysis. It seems like there is two months' worth of data from this project, so what makes 02-July-2016 so unique that it warrants its own paper, and how representative is it of typical flow patterns for this regime in the region?

The flight made in the afternoon of 2 July is unique in the sense that it is the only flight conducted over the ocean during which the downward looking lidar ULICE was operational. The combination of remote sensing to monitor the aerosol landscape over the Gulf of Guinea and in situ measurements to assess the nature of the observed aerosols was only possible on that day. For information, two other so-called OLACTA flights were conducted with the ATR 42 during the campaign. However, the lidar was not working and only in situ, low-level measurements were made.

We have added this bit of information in the Introduction, at the end of the penultimate paragraph as:

"The flight made in the afternoon of 2 July is unique in the sense that it is the only flight conducted over the ocean during which a downward looking lidar was operational. The combination of remote sensing to monitor the aerosol landscape over the Gulf of Guinea and in situ measurements to assess the nature of the observed aerosols was only possible on that day." The primary concerns raised in the specific comments section regard aerosol aging and water uptake in humid environments, and timing of tracer release and the interpretation of maximum aerosol extent in the model.

We have hopefully clarified these issues in the following.

It is recommended that the manuscript be published in ACP after the specific and technical comments are addressed in the paper.

**Specific Comments**

Page 2 - Lines 32-34) States that the lower troposphere aerosol loading includes emissions from Lagos, but later in the paper the tracer experiment shows that the aerosol plumes over the ocean do not have a signal from Lagos. Is this a reference to Lagos being an aerosol source in the SWA region, instead of the over the limited ocean aircraft data from this case study?

Absolutely. Lagos is a large source of anthropogenic emissions in SWA. However in the domain of operation of the aircraft, which is quite far west compared to Lagos, and given the general direction of the monsoon flow, emissions from Lagos did not impact air quality in the region of interest for the present study.

We have modified the sentence in lines 41-42 (abstract) and lines 814-816 (conclusion) to clarify this in the revised manuscript.

In the abstract:

"Given the general direction of the monsoon flow, the tracer experiments indicate no contribution from Lagos emissions to the atmospheric composition of the area west of Cotonou, where our airborne observations were gathered."

**In the conclusion:**

"[...] given the general direction of the monsoon flow, Lagos emissions (taken to be 13 times that of Cotonou) do not appear to have affected the atmospheric composition west of Cotonou, where our airborne observations were gathered, as also shown by Deroubaix et al. (2018) in the summer in post-monsoon onset conditions, [...]"

Page 4 - Lines 81-84) Is the purpose of DACCIWA / this paper to understand how atmospheric dynamics influences aerosol emission rates (e.g. stronger surface winds will loft more dust), or only aerosol transport after emission, or both?

The purpose of DACCIWA is to understand aerosol transport after emission.

**The sentence was modified in the revised manuscript to include this information:**

"One of the aims of the EU-funded project Dynamics-Aerosol-Chemistry-Cloud Interactions in West Africa (DACCIWA, Knippertz et al., 2015b) is to understand the influence of atmospheric dynamics on the spatial distribution of both anthropogenic and natural aerosols over SWA after emission." Page 7 - Lines 161-163) Even though the optical properties could not be retrieved with the ULICE lidar inversion procedures, they were retrieved using other instrumentation, correct? If not, what was excluded?

Yes, the optical properties could be retrieved with other instrumentation described in the subsequent sub-section (section 2.1.2), namely a Particle Soot Absorption Photometer, a CAPS-PMex, and an integrated nephelometer (Ecotech, model Aurora 3000).

Pages 7-8 - Lines 178-186) Can sea salt be identified with this method?

No, the selected gas phase chemistry and aerosol metrics selected are only intended to discriminate between biomass burning aerosols associated with long-range transport from the south, anthropogenic pollution and dust particles associated with long-range transport from the north. In order to formally identify sea-salt we would need to include filter analysis, which were not conducted at this point. Nevertheless, sea salt particles can be identified from the lidar measurements as being associated with high backscatter and low depolarization (as discussed in the paper) as well as reflected in the large particles concentration (NPM2.5) measured over the ocean and inland. However, from this we are not able to segregate sea-salt from other aerosols in case of mixture in the ABL.

A sentence was added in Section 2.1.2 in the revised manuscript, after the description of the metrics:

"Sea salt cannot formally be identified with the in situ measurements conducted with the ATR 42 payload during DACCIWA."

Pages 7-8 - Lines 178-186) What happens when there is a mixture of aerosol species instead of homogeneous plumes? Looking at Figure 10-b, the CALIOP data suggests a heterogeneous aerosol air mass during this case study event (e.g. dust mixing with smoke).

We agree that homogeneous plumes for a given aerosol type will likely only be observed fairly close to the sources, and that in the broader area of the aircraft operation, mixing is likely to occur. Rather than indicating homogeneous plumes, our metrics are an indication of what type of aerosol dominates the composition of a given sampled air mass. This is now more clearly stated in the revised manuscript.

A couple of sentences were added at the end of the 1st paragraph of Section 2.1.2 in the revised manuscript:

"Because of the complex atmospheric dynamics in the area, we cannot assume that only homogeneous air masses will be sampled with the aircraft. Rather, the selected observations are indicators of which type of aerosol dominates the composition of a given sampled air mass."

Pages 7-8 - Lines 178-186) Because the aircraft measurements were taken over the ocean, the particles reside in a relatively humid atmosphere. Depending on the aerosol species and the humidity of the environment, particles can take up water, changing their diameter and their optical properties. Does this affect VDR values or any metric by which the aerosol species were partitioned? Would it change the

analysis in later sections at all, especially pertaining to attribution of fresh versus aged plumes?

VDR values of large non-spherical particles will be affected by humidity, in the sense that water absorption will make these particles more spherical, and hence decrease the associated VDR values. On the other hand, small pollution particles (local anthropogenic or resulting from biomass burning far south) generally do not depolarize much, at least not at the wavelength of the lidar. Therefore, the VDR value of pollution particles having taken up water will not be significantly modified (i.e. will remain within the uncertainty of the VDR retrieval method).

From the in-situ perspective, relative humidity might indeed affect some of the measurement properties (as correctly pointed out by the reviewer, the optical properties of more hygroscopic components of aerosols, for example). However, most of the aerosol sampling lines are heated (to 35-40°C), effectively limiting water uptake and relative humidity to values below 40%. Therefore aerosol properties derived from in-situ measurements are given for dry conditions.

Furthermore, the goal here is to obtain a general classification into aerosol types, achieved via a combination of collocated metrics, most of which (e.g. gas-phase, or total aerosol number > 10 nm) are typically insensitive to relative humidity. Therefore, the effect of aerosol water uptake is not considered to be a source of bias in the analysis presented here.

A sentence was added on this. See answer to next point.

Page 8 – Lines 181-182) What about urban O3? Will that mislead the speciation between smoke and pollution?

The  $O_3$  measurements in the ATR are based on dual cell technology (a Thermo Environmental Instrument – TEI 49), and therefore largely insensitive to ambient relative humidity according to Spicer et al. (2010).

Spicer, C. W., D. W. Joseph and W. M. Ollison, 2010: A Re-Examination of Ambient Air zone Monitor Interferences, J. Air & Waste Manage. Assoc. **60**:1353–1364.

A couple of sentences were added after the description of the aerosol types (Section 2.1.2) in the revised manuscript to cover this point and the previous one: "Gas phase and aerosol metrics above are typically insensitive to relative humidity. The aerosol sampling lines are heated (to 35-40°C), effectively limiting water uptake and relative humidity to values below 40%. The O3 measurements in the ATR are based on dual cell technology, and therefore largely insensitive to ambient relative humidity according to Spicer et al. (2010), in spite of the humid environmental conditions over the Gulf of Guinea."

Page 8 – Line 202) It is stated that "data were processed with a time resolution of 1 s" – is this for all data or just the CAPS-Mex data? Was there some standard time resolution used for interpolation across instrumentations to line up the time resolutions? If so, what interpolation technique was used? 1 s resolution is for the CAPS-Mex data in that case. We have used the native resolution of the instrument or have averaged measurements to a coarser resolution, as indicated in Table 1 (note that we have completed Table 1 where this information was lacking). We have not attempted to line up the time evolution of the different instruments and therefore have not used any interpolation technique to plot the data.

**Page 13 – Lines 322-324) Is this one-way or two-way nesting in WRF?**

WRF is used to compute the meteorology and CHIMERE for the transport of chemical species and tracers. The CHIMERE model is forced off-line by WRF. The WRF simulations are performed before CHIMERE and independently of the species to transport. For WRF, it is two-ways nesting and for CHIMERE it is one-way nesting.

This information has been added in the revised manuscript (see reply to the subsequent comment).

Page 13 – Lines 326-327) More description of the WRF setup and physics options is necessary, especially the PBL parameterization, since the WRF PBL height is used later on in the paper. Furthermore, the WRF parameterizations used generally get a reference citation. Does the statement that the model configuration is the same as in Deroubaix et al. 2018 mean that every physics option / parameterization is identical to their setup? What about time steps, output intervals, and nudging? The Deroubaix et al., 2018 simulation was for a similar region in SWA, but the grid spacing was coarser, the simulation was run for a much longer duration to study short-term climate phenomena, and they ran with active chemistry instead of tracers. Stating that the setup is the same as in Deroubaix et al. 2018 may be confusing when these differences are considered.

The WRF set-up is strictly the same as the one fully described in Deroubaix et al. (2018), except of the grid spacing. The description of the schemes used in WRF was not included again in the present paper because the differences in resolution and duration have no impact on the choice of physics parameterizations. The fact that CHIMERE is running active chemistry or passive tracers is also independent of the choices made to calculate the meteorology with WRF. CHIMERE being used in off-line mode, the meteorology is calculated first, before the start of the CHIMERE simulation.

Nevertheless, for the sake of clarity and self-coherence, the text was changed and now reads:

"The WRF model (version v3.7.1, Shamarock and Klemp, 2008) and the CHIMERE chemistry-transport model (2017 version, Mailler et al., 2017) are used in this study. WRF calculates meteorological fields that are then used in off-line mode by CHIMERE to (i) conduct tracer experiments and (ii) compute backplumes. WRF and CHIMERE simulations are performed on common horizontal domains and with the same horizontal resolution. For the period 30 June--3 July 2016, two simulations are conducted for both WRF and CHIMERE to provide insights into the airborne observations: a simulation with a 10-km mesh size in a domain extending from 1°S to 14°N and from 11°W to 11°E (larger than the domain shown in Figure 1a) and a simulation with a 2-km mesh size in a domain extending from 2.8°N to 9.3°N and from 2.8°W to 3.3°E (Figure 1a).

The nested WRF simulations are first performed with hourly outputs. For the two horizontal resolutions, the same physical parameterizations are used and are those described in Deroubaix et al. (2018). The ABL scheme is the one proposed by the Yonsei University (Hong et al., 2006), the microphysics is calculated using the Single Moment-6 class scheme (Hong and Lim, 2006), the radiation scheme is RRTMG (Mlawer et al., 1997), the cumulus parameterization is the Grell-Dévényi scheme and the surface fluxes are calculated using the Noah scheme (Ek et al., 2003). The 10-km WRF simulation uses National Centers for Environmental Prediction (NCEP) Final global analyses as initial and boundary conditions. NCEP Real-Time Global SSTs (Thiébaux et al., 2003) are used as lower boundary conditions over the ocean. The meteorological initial and boundary conditions for the 2-km WRF simulation are provided by the 10-km WRF run, which, in turn, receives information from the 2-km WRF simulation (two-way nesting). The simulations are carried out using 32 vertical sigma-pressure levels from the surface to 50 hPa, with 6 to 8 levels in the ABL.

Then the CHIMERE simulations are performed. The horizontal grid is the same as for the lower resolution WRF runs. Vertically, CHIMERE uses 20 levels from the surface to 300 hPa and three-dimensional meteorological fields are vertically interpolated from the WRF to the CHIMERE grid. The two-dimensional fields, such as 10-m wind speed, 2m temperature, surface fluxes and boundary-layer height are used directly in CHIMERE. The chemistry and aerosol initial and boundary conditions for the 2-km CHIMERE simulation are provided by the 10-km simulation (one-way nesting)."

Ek, M. B., Mitchell, K. E., Lin, Y., Rogers, E., Grunmann, P., Koren, V., Gayno, G., and Tarpley, J. D., 2003: Implementation of Noah land surface model advances in the National Centers for Environmental Prediction operational mesoscale Eta model, J. Geophys. Res.-Atmos., 108, 8851.

Hong, S. and Lim, J., 2006: The WRF single-moment 6-class microphysics scheme (WSM6), 42, 129–151.

Hong, S.-Y., Noh, Y., and Dudhia, J., 2006: A new vertical diffusion package with an explicit treatment of entrainment processes, Mon. Weather Rev., 134, 2318–2341.

Mlawer, E. J., Taubman, S. J., Brown, P. D., Iacono, M. J., and Clough, S. A., 1997: Radiative transfer for inhomogeneous atmospheres: RRTM, a validated correlated-k model for the longwave, J. Geophys. Res., 102, 16 663.

Page 14 – Lines 344-346) Is there any observational evidence or prior literature that supports scaling urban emissions by population in this way? For example, why couldn't an efficient metropolis have 5x the population as a baseline city, but only 2x the pollution? Does the linear scaling of population and pollution break down at some point for this region or other regions?

We agree with the reviewer: efficient megacities may have 5x the population compared to a 'baseline city' but only 2x the pollution. However, large cities of developing countries in West Africa are known not to be 'efficient' due to a lack of adequate policies. Here, our goal is to use tracers in CHIMERE to look at the spatiotemporal structure of city plumes, away from emissions and after transport. Considering that African cities generate an atmospheric pollution roughly proportional to their total population is as good a first guess as any. Furthermore, the differences in emissions scaled to the population for the cities of Accra and Lomé are not so different from Cotonou (3x and 1.8x, respectively), unlike Lagos (13x). However, Lagos emissions did not impact the air quality over the area of interest for this case study, as explained in the manuscript. Hence, even in the event that emissions are not strictly proportional to city population and that the 3x and 1.8x factors were slightly different, the conclusion drawn from the tracer experiments would not be changed.

Our approach would have been different if we wanted to relate a maximum of concentration observed with the aircraft over a city. In such a case, we would need to consider emissions density and then population density, not total population.

A sentence was added in the revised version of the manuscript:

"Large cities in developing countries are generally considered to generate an atmospheric pollution roughly proportional to their total population due to a lack of adequate emission policies."

Page 14 – Lines 347-349) The naming of the simulations is a bit counterintuitive. Instinctively, I'd think that TRA\_D1 would represent July 1st and TRA\_D2 as July 2nd. However, TRA\_D2 is July 1st and TRA\_D3 is July 2nd. By the time these simulations were discussed 13 pages later, the numbering became confusing. Perhaps numbering related to the dates would help readers later on (e.g. TRA\_D12 = July 1st-2nd, TRA\_D1= July 1st only, TRA\_D2 = July 2nd only).

Agreed. We have modified the denomination of the experiments as suggested. Furthermore, experiment TRA\_II was renamed TRA\_II2 to be coherent with the naming of experiments TRA\_Dx.

Page 14 – Lines 353-355) What does it mean that the lifetime of the tracers is designed to be 48 hours? Why set the concentration to zero if they are still present in the domain after 48 hours? Is it because the tracers do not undergo gravitational settling? Would including the gravitational settling process change the interpretation in later sections?

Sorry about the confusion here. The mention to a 48 h lifetime and setting concentrations to zero after that time is erroneous. This set up corresponds to previous model configurations, not the one used in this study and described in Mailler et al. (2017). The tracers are continuously emitted and there is no lifetime. The sentence, lines 353-355, was completely removed. About the settling, this process is not taken into account for the tracers as they are considered as 'gaseous' tracers.

Page 15 – Line 365) Why are the tracers released at 2500 m ASL?

This is based on the altitude of the elevated biomass burning layer arriving from the south (feature E seen in the Figure 3a). Since this information is provided later, we have added a sentence here to justify this.

The following sentences have been added in the revised manuscript (2nd and 4th sentences of Section 3.2.2):

"The objective is to assess the origin of an elevated aerosol layer observed with the lidar ULICE (see Section 5)."

"For both locations, backplumes are launched at 2500 m above sea level on 2 July 2016 at 17:00 UTC (i.e. the height of the elevated aerosol layer above the Gulf of Guinea, see Section 5)."

Page 18 – Lines 462-465) Maybe the placement of the 'A' on Figure 3 is misleading. To me, it looks like the 'A' is pointing to shallow clouds and not an aerosol layer.

We have added an arrow in Figure 3, to point to 'A' to make things clearer.

Mention to the added arrow is now made in the caption of Figure 3.

Page 20 – Line 503) Is there an explanation for why there is a reduction in O3 concentrations compared to background levels for Plume A?

Plume A is related to fresh anthropogenic emissions from Lomé, including NOx. The addition of a large quantity of NOx into the atmosphere can lead to a significant shift in the ozone chemical equilibrium, which can effectively result in near-source consumption, as observed here.

The following has been added in the revised version of the manuscript:

"[...] together with an  $O_3$  concentration reduction (Figure 5b). Plume A is related to fresh anthropogenic emissions from Lomé, including NOx. The addition of a large quantity of NOx into the atmosphere can lead to a significant shift in the ozone chemical equilibrium, which can effectively result in near-source consumption, as observed here."

Page 20 – Line 506) What is the significance of the O3 to CO ratio? Why does the value of 0.15 imply the plume is fresh versus a value of 0.25 implies that it is aged?

The  $O_3/CO$  ratio is an indicator of the aging of air mass during transport. Whereas the actual  $O_3/CO$  ratio depends on a number of parameters, such as background CO, source emission profile, insolation, availability of  $O_3$  precursors, atmospheric reactivity, etc..., to the first order the ratio increases as the plume is aging (e.g. Jaffe and Wigder, 2012, and Kim et al., 2013). This is because, in the troposphere, the ozone production continues as long as  $NO_x$  is available, whereas CO concentrations decrease slightly during transport. Hence, the actual increase of this ratio by 65% observed here is more meaningful than the values itself. To reflect this more clearly, the sentence on P.20 L.506 has been removed and P.20 L.517 has been modified to now read:

"The  $O_3/CO$  ratio (an indicator of air mass aging, e.g. Jaffe and Wigder (2012) and Kim et al. (2013)) observed to be associated with feature B increases with respect to feature A (0.25 vs. 0.15, i.e. a 65% increase), which is compatible with a further processed urban plume, as also corroborated by wind measurements. "

Jaffe, D. A. and N. L. Wigder, 2012: Ozone production from wildfires: A critical review, Atmos. Env., 51, 1-10.

Kim, P. S., D. J. Jacob, X. Liu, J. X. Warner, K. Yang, K. Chance, V. Thouret and P. Nedelec, 2013: Global ozone–CO correlations from OMI and AIRS: constraints on tropospheric ozone sources, Atmos. Chem. Phys., 13, 9321–9335.

Page 23 – Lines 592-593) This is regarding the statement that the emissions come only from July 1st. Figure 4 shows the wind speeds above 500 m to be weak (1-2 m/s), so the emissions on July 2nd haven't had a chance to be advected far from their source regions in the weak winds. It makes sense then that the emissions must be from July 1st, or an earlier date. Is it possible that due to the low wind speeds above the PBL that what we are seeing isn't just from July 1st, but also June 30th? Would the picture change if the tracers were released starting on June 30th?

If we compare Figure 7a (TRA\_D12, new nomenclature proposed by the referee, previously TRA\_D1) and 7d (TRA\_D2, new nomenclature, previously TRA\_D3), it is clear that the difference is related to emissions on 1 July and that the differences are observed above the marine ABL, in the region of the easterly flow (centered at ~1.5 km amsl) where the winds are not so weak. It is fair to say that emission from the 30 June will contribute to the overall picture, however, given the proximity of the western boundary of the 2-km CHIMERE domain to the western part of the aircraft flight track, we are confident that the tracers from 30 June would have been advected out of the domain in the afternoon of 2 July.

Page 26 – Lines 671-675) Do you think the maximum extent that the plume reaches over the ocean in the model is related to the tracer lifetime and the end time of the simulation? If the simulation was run for longer, would the maximum tracer extent over ocean increase? This goes back to the previous comment about releasing tracers on June 30th. If the tracers have no settling velocity or cannot be scavenged by precipitation, they could be advected indefinitely in the model.

There is no fixed lifetime for the tracers as explained above. We do acknowledge that this was not clear in the original version of the manuscript and it is only fair that the reviewer inquiries about this given the elements provided at the time.

The extent of the plume is mainly controlled by the direction of the mid-level easterly winds (and the small northerly component associated with it), as explained in Section 6.2.

We have re-emphasized this in the Conclusion by modifying the last sentence of the antepenultimate paragraph:

"[...]and (d) the tracer plumes do not extend very far over the ocean during the short period under scrutiny, mostly because they are transported northward within the marine ABL and westward above it so that their extent is controlled by the equatorward component in the mostly easterly flow as modulated by the synoptic-scale disturbances (Knippertz et al., 2017)."

Furthermore, when looking at the meridional wind extracted over Accra over the months of June and July 2016 (see Figure to the left), we observe that at the mean altitude of the easterly flow above the monsoon flow (~800 hPa) there is an alternation of northerly and southerly components imposed by the propagation of African Easterly Waves. This alternation is really what limits the extent of the pollution plume over the ocean, as the meridional component changes from northerly to southerly every ~3 days during the 2 months.

Page 30 – Lines 751-752) Why is the correlation here related to terrain? I'm not sure I see the connection between skin temperature, vertical velocity, and terrain.

The meridional gradient of skin temperature between the sea and the land is an indicator for the pressure difference and thus drives the intensity of the southerly flow associated with the land sea breeze. When the southerly flow impinges on the low terrain over SWA, as it progresses over the continent, enhanced vertical motion is generated.

This information has been added in the revised version of the manuscript.

Page 35 – Lines 897-905) Was the flight over the Mediterranean an aerosol-free environment for calibration? If not, how might that affect the accuracy or uncertainty in the retrievals?

The ATR flight over the Mediterranean was conducted from an altitude above 6 km amsl, with ULICE lidar data acquired between 0 and 6 km amsl (see Figure below). The calibration was performed using lidar data acquired around 1528 UTC well above the aerosol layer, i.e. between 5 and 6 km amsl where the lidar backscatter is only sensitive to the molecular background signal.

---

## Editor Decision (ED1)

acp-2018-346
Aerosol distribution in the northern Gulf of Guinea: local anthropogenic sources, long-range transport and the role of coastal shallow circulations

Dear Dr. Flamant,

I have reviewed your manuscript revisions, as well as your responses to the concerns raised by both reviewers.

It is clear that you have worked hard to address all of the reviewer comments, both within the manuscript and in your responses. I am satisfied with all of your responses to both reviewers, with the exception of the primary concern raised by referee #2 regarding the generalization of the results. Referee #1 also touched in this in their question regarding what makes this particular day worthy of analysis.

I read through the manuscript and found a number of places where the extent of generalization could, and indeed should, be toned down using words like "could", "may", and so on. More particularly, I think that the Abstract, Introduction and the Conclusion should be edited to reflect this. The following are some examples of what should be considered. I would encourage the authors to address whether there are any others.

(1) Abstract
- The last sentence should include "can" before distribute. While your analysis is certainly accurate for this particular day, and while the role of these flow regimes has not been documented, the fact remains that this is only one possible way in which this may happen.
- I would include a statement regarding how representative this day is of the typical meteorological situation in July. Your responses indicate that you have performed this analysis, and you mention it in the manuscript. I would therefore allude to this in the abstract.
- After "Ghana and Togo" it is recommended that you include a statement as to how typical this flow regime is.
- Include a statement as to why this day was unique in terms of the lidar being operationa.

(2) Introduction
- Line 106: "The main objective of the present study is to understand how the lower tropospheric circulation over SWA shapes the urban pollution plumes emitted from coastal cities …." The current study contributes to this, however, only for one particular regime. There are likely to be many. It is recommended that something like "can shape" or "one of the mechanisms by which the lower tropospheric circulation … can shape"
- Again, it would be useful to comment on how common the synoptic setup is for this region.
- Some statement should be made regarding that this is a study of only one day and that caution should be exercised when drawing more general conclusions regarding the role of observed circulation in aerosol redistribution in this region.

(3) Conclusion
- Similar statements as noted above for the Abstract and the Introduction should be included in the Conclusion.

I look forward to reading your revised version that takes into account these suggestions.

Kind regards,
Sue van den Heever

---

## Author Response (AR2)

**Aerosol distribution in the northern Gulf of Guinea: local anthropogenic sources, long-range transport and the role of coastal shallow circulations**

**By C. Flamant et al.**

Reply to the co-Editor's comments

In the following, the comments made by the co-Editor appear in black, while our replies are in red, and the proposed modified text in the typescript is in blue.

Co-Editor's comments

I have reviewed your manuscript revisions, as well as your responses to the concerns raised by both reviewers. It is clear that you have worked hard to address all of the reviewer comments, both within the manuscript and in your responses. I am satisfied with all of your responses to both reviewers, with the exception of the primary concern raised by referee #2 regarding the generalization of the results. Referee #1 also touched in this in their question regarding what makes this particular day worthy of analysis. I read through the manuscript and found a number of places where the extent of generalization could, and indeed should, be toned down using words like "could", "may", and so on. More particularly, I think that the Abstract, Introduction and the Conclusion should be edited to reflect this. The following are some examples of what should be considered. I would encourage the authors to address whether there are any others.

We would like to thank the co-Editor for her encouraging comments on the revised version of the paper, as well as helpful suggestions on how to further improve the paper. We have gone through the Abstract, Introduction and Conclusion to modify the paper and comply with the co-Editor's demand.

(1) Abstract
• The last sentence should include "can" before distribute. While your analysis is certainly accurate for this particular day, and while the role of these flow regimes has not been documented, the fact remains that this is only one possible way in which this may happen.

Agreed. We have modified the sentence accordingly.

The sentence now reads:
This work sheds light on the complex – and to date undocumented – mechanisms by which coastal shallow circulations can distribute atmospheric pollutants over the densely populated SWA region.

• After "Ghana and Togo" it is recommended that you include a statement as to how typical this flow regime is.

Agreed. We have added this information.

The sentence now reads:
Our results indicate that the aerosol distribution on this day is impacted by subsidence associated with zonal and meridional regional-scale overturning circulations associated with the land-sea surface temperature contrast and orography over Ghana and Togo, as typically observed on hot, cloud-free summer days such as 2 July 2016.

• I would include a statement regarding how representative this day is of the typical meteorological situation in July. Your responses indicate that you have performed this analysis, and you mention it in the manuscript. I would therefore allude to this in the abstract.

Agreed. We have conducted an analysis of the occurrence of the zonal circulation we have in the course of July 2016. As stated in the manuscript, the zonal circulation is a general feature of July 2016 and not only unique to the 2 July 2016 case. We have added this information in the abstract.

The following sentence was added after the sentence modified in the previous comment (i.e. the addition to the sentence ending with "[…] Ghana and Togo"): "Furthermore, we show that the zonal circulation evidenced on 2 July is a persistent feature over the Gulf of Guinea during July 2016."

• Include a statement as to why this day was unique in terms of the lidar being operational.

Agreed. We have added a sentence to reflect this in the abstract.

The following sentence was added after the 2nd sentence of the abstract: "This was the only flight conducted over the ocean during which a downward looking lidar was operational."

(2) Introduction
• Line 106: "The main objective of the present study is to understand how the lower tropospheric circulation over SWA shapes the urban pollution plumes emitted from coastal cities …." The current study contributes to this, however, only for one particular regime. There are likely to be many. It is recommended that something like "can shape" or "one of the mechanisms by which the lower tropospheric circulation … can shape".

Agreed. We have modified the sentence to reflect this.

The modified sentence now reads:
"The main objective of the present study is to understand how shallow overturning circulations developing in the lower troposphere over SWA on hot, cloud-free days can shape the urban pollution plumes […]"

• Again, it would be useful to comment on how common the synoptic setup is for this region.

Agreed. We have added a sentence to reflect this.

The following sentence was added after the 4th sentence of the penultimate paragraph of the Introduction:

"We show that the aerosol distribution on this day is impacted by subsidence associated with zonal and meridional regional-scale overturning circulations linked with land-sea surface temperature contrast and orography over Ghana and Togo, and that the zonal circulation evidenced on 2 July is a persistent feature over the Gulf of Guinea during July 2016."

• Some statement should be made regarding that this is a study of only one day and that caution should be exercised when drawing more general conclusions regarding the role of observed circulation in aerosol redistribution in this region.

Agreed. We have added such a statement near the end of the Introduction, before the paragraph detailing the content of the different sections.

The following sentence has been added at the end of the penultimate paragraph: "Therefore, one should keep in mind that we are detailing a few mechanisms possibly responsible for shaping the aerosol composition over the Gulf of Guinea, and caution should be exercised when drawing more general conclusions regarding the role of observed circulation in aerosol redistribution in this region."

(3) Conclusion
• Similar statements as noted above for the Abstract and the Introduction should be included in the Conclusion.

Agreed. We have made a series of modifications to further account for the above mentioned statements. Please note that some of these aspects have already been dealt with in the previously edited revised version.

The modifications to the "Summary and conclusions" Section are listed below. Line numbers refer to the "track change" version of the typescript:
  o Line 867: "[…] on that day" was added,
  o Line 905: "[…], on hot, cloud-free summer days such as 2 July, […]"
  o Line 913: we have added a cautionary statement at the end of the 3rd paragraph: "Still, one should keep in mind that the mechanisms described in details are based on a unique dataset. Even though we have highlighted the fact that some of the key dynamical features are persistent during July 2016, and hence not just representative of 2 July, caution should be exercised when drawing more general conclusions regarding the role of observed circulation in aerosol redistribution in this region."

We hope that the above changes meet your expectations.

Best regards

Cyrille Flamant, on behalf of all co-authors

[revised manuscript text omitted]